# Fungal small RNAs ride in extracellular vesicles to enter plant cells through clathrin-mediated endocytosis

Baoye He[1], Huan Wang[1], Guosheng Liu[2], Angela Chen [1], Alejandra Calvo[1], Qiang Cai [2] & Hailing Jin [1] ✉

Small RNAs (sRNAs) of the fungal pathogen *Botrytis cinerea* can enter plant cells and hijack host Argonaute protein 1 (AGO1) to silence host immunity genes. However, the mechanism by which these fungal sRNAs are secreted and enter host cells remains unclear. Here, we demonstrate that *B. cinerea* utilizes extracellular vesicles (EVs) to secrete Bc-sRNAs, which are then internalized by plant cells through clathrin-mediated endocytosis (CME). The *B. cinerea* tetraspanin protein, Punchless 1 (BcPLS1), serves as an EV biomarker and plays an essential role in fungal pathogenicity. We observe numerous *Arabidopsis* clathrin-coated vesicles (CCVs) around *B. cinerea* infection sites and the colocalization of *B. cinerea* EV marker BcPLS1 and *Arabidopsis CLATHRIN LIGHT CHAIN 1*, one of the core components of CCV. Meanwhile, BcPLS1 and the *B. cinerea*-secreted sRNAs are detected in purified CCVs after infection. *Arabidopsis* knockout mutants and inducible dominant-negative mutants of key components of the CME pathway exhibit increased resistance to *B. cinerea* infection. Furthermore, Bc-sRNA loading into *Arabidopsis* AGO1 and host target gene suppression are attenuated in those CME mutants. Together, our results demonstrate that fungi secrete sRNAs via EVs, which then enter host plant cells mainly through CME.

Communication between hosts and microbes/parasites involves the exchange of bioactive molecules, such as proteins and metabolites. Recent studies show that RNA molecules, particularly regulatory small RNAs (sRNAs), also travel between hosts and interacting organisms to induce gene silencing in *trans*, a mechanism termed "cross-kingdom RNA interference" (RNAi)[1–14]. Since the initial discovery of cross-kingdom RNA trafficking[3], this regulatory mechanism has subsequently been observed between plants and many fungal pathogens[11,12,15–17], oomycete pathogens[13], and parasitic plants[14]. Interestingly, cross-kingdom sRNA trafficking has been found to be important even for symbiosis, specifically in mycorrhizal interactions[10] and between prokaryotic Rhizobia and host soybean roots[8]. This mechanism is not limited to plant-microbe/parasite interactions and has also been observed in animal-pathogen/parasite systems[4–6]. In

many of these interactions, the transferred microbial sRNAs can utilize host AGO proteins to silence host immune response genes[3,8,11–13]. Given the importance of these sRNA effectors in pathogen virulence, antifungal strategies that target and silence genes in the RNAi machinery, such as the Dicer-like (DCL) proteins that generate sRNAs, have been shown to be effective for plant protection[16,18–22]. Further protection could likely be achieved by inhibiting the transport processes that enable cross-kingdom trafficking of these sRNAs. However, how these transferred sRNAs are delivered and internalized by plant cells is still a mystery.

Extracellular vesicles (EVs) are used across the kingdoms of life to package and deliver functional molecules into recipient cells or organisms[2,6,23–26]. These EVs can provide protection to their cargos from degradation by extracellular enzymes, which is especially

[1]Department of Microbiology and Plant Pathology, Center for Plant Cell Biology, Institute for Integrative Genome Biology, University of California, Riverside, CA, USA. [2]State Key Laboratory of Hybrid Rice, College of Life Science, Wuhan University, Wuhan, China. ✉e-mail: hailingj@ucr.edu

important for RNA trafficking. Indeed, EVs have been utilized by plants to deliver plant small RNAs to fungal pathogens during cross-kingdom RNAi[2]. Furthermore, many fungal species have been shown to secrete EVs, including yeast and filamentous fungi[27–30]. Although, only recently have EVs from plant pathogenic fungi such as the maize smut pathogen *Ustilago maydis*[31], the wheat pathogen *Zymoseptoria tritici*[32], and the cotton pathogen *Fusarium oxysporum f.* sp. *vasinfectum*[33] been characterized. Still, it remains unclear whether plant fungal pathogens secrete EVs to deliver sRNAs into interacting plant cells.

Endocytosis is the primary route for eukaryotic cells to internalize plasma membrane proteins, lipids, and extracellular materials into the cytoplasm[34,35]. In plants, endocytosis is involved in many critical biological processes, including growth, development, signal transduction, nutrient uptake, and plant-microbe interactions[36–39]. In the past decades, substantial progress has been made in characterizing the function of endocytosis in plant immune responses. For instance, several plasma-membrane localized receptor proteins involved in the plant defense response are internalized through the endocytosis pathway[36,39–41]. Analogous to animal and yeast cells, clathrin-mediated endocytosis (CME) is the main route to internalize extracellular and plasma membrane materials in plants[42]. Clathrin is a triskelion-shaped complex composed of three clathrin heavy chains (CHCs) and three clathrin light chains (CLCs)[34]. By interacting with the plasma membrane, cargo proteins, adaptor protein 2 complex (composed of α, β2, μ2, and σ2 subunits) and other accessory protein factors, clathrin-coated pits (CCPs) can be formed[34,43]. The CCPs will then bud off from the plasma membrane to form clathrin-coated vesicles (CCVs) to deliver cargos into cells[34]. CME also contributes to the internalization of protein effectors from the fungal pathogen *Magnaporthe oryzae* and the oomycete pathogen *Phytophthora infestans* into plant cells[44,45]. In addition to CME, other sterol-sensitive clathrin-independent endocytosis (CIE) pathways exist in plants. The most well-studied proteins in the CIE pathway are membrane microdomain-associated flotillin1 and remorins[46,47]. Whether CME and/or CIE pathways play a role in the internalization of fungal sRNA effectors and EVs is unknown.

Here, we find that *B. cinerea* secretes EVs to deliver fungal sRNAs, and the *B. cinerea* tetraspanin protein BcPLS1 is essential for *B. cinerea* pathogenicity and EV-mediated Bc-sRNA secretion. Furthermore, utilizing genetics, biochemistry, and cell biology approaches, we demonstrate that CME is the major pathway for the uptake of fungal EVs and their sRNA cargo.

## Results
### *B. cinerea* can secrete BcPLS1-positive EVs during infection

*B. cinerea* EVs were first isolated from the supernatants of wild-type strain B05.10 liquid culture using the traditional differential ultracentrifugation (DUC) method[27]. Nanoparticle tracking analysis (NTA) was used to determine the size distribution of isolated EVs, and a peak at 113 nm was observed for *B. cinerea* EVs (Supplementary Fig. 1a). The isolated EVs were further examined by transmission electron microscopy (TEM), which confirmed the typical cup-shaped form of EVs (Supplementary Fig. 1b). To further purify EVs obtained from *B. cinerea* liquid culture, the isolated EVs were subjected to sucrose gradient centrifugation as per the minimal information guidelines for studies of extracellular vesicles 2018 (MISEV2018)[48]. Analysis using NTA indicated that EVs were predominantly concentrated in sucrose fractions four and five with peak diameters of 93 nm and 94 nm, respectively (Fig. 1a). EVs were observed in both fractions and exhibited typical cup-shaped structures as confirmed by negative staining and TEM analysis (Fig. 1b). The morphology and size distribution of the isolated *B. cinerea* EVs were comparable to EVs from other species, including plants and plant pathogenic fungi[31–33,49], suggesting that *B. cinerea* has the ability to secrete EVs into the external environment.

Some tetraspanin proteins, such as CD63 in mammalian cells and tetraspanin 8 (TET8) in plant cells, are known to be markers for EVs[2,50].

In *B. cinerea*, two tetraspanin coding genes, Punchless 1 (*BcPLS1*) and Tetraspanin 3 (*BcTSP3*), have been discovered[51]. Although these two proteins display limited amino acid sequence similarity, they share conserved structural hallmarks, including four transmembrane domains (TM1 to TM4), a small extracellular loop (ECL1), an intracellular loop (ICL), and a large extracellular loop (ECL2) (Supplementary Fig. 2a, b). To determine whether these tetraspanin proteins are associated with *B. cinerea* EVs, both BcPLS1 and BcTSP3 proteins were tagged with a fluorescent protein YFP and inserted into the *B. cinerea* genome by homologous recombination. The EVs were isolated and separated using sucrose gradients from the liquid culture of both transgenic strains. As shown in Fig. 1c, BcPLS1 was found in the EV fractions three to five at the density of 1.11–1.19 g ml⁻¹ and mainly concentrated at fraction four. However, BcTSP3 was hardly detected in isolated EV fractions, although it can be expressed in *B. cinerea* cells (Fig. 1c and Supplementary Fig. 2c). Meanwhile, BcPLS1-YFP-positive EV signals can be observed under confocal microscopy (Fig. 1d).

Both transformed strains were then used to challenge wild-type *Arabidopsis* plants and samples were examined by confocal microscopy ten hours post-inoculation. As shown in Fig. 1e, BcPLS1-YFP-labeled vesicles were clearly observed outside of the fungal cells around infection sites. However, it was difficult to detect secreted vesicle signals from BcTSP3-YFP-infected plants. EVs were then isolated from BcPLS1-YFP or BcTSP3-YFP infected plants and separated using sucrose gradients. Both BcPLS1-YFP and *Arabidopsis* exosome marker TET8 were concentrated in the same fractions (Fig. 1f). To further confirm that *B. cinerea* secretes EVs during infection, BcPLS1-YFP or BcTSP3-YFP strains were used to inoculate *Arabidopsis* TET8-CFP transgenic plants. EVs were isolated from the apoplastic fluid of infected plants using sucrose gradients, and fraction four was examined under confocal microscopy. The EVs from BcPLS1-YFP-infected TET8-CFP plants exhibited two distinct fluorescent protein-labeled signals (Fig. 1g). These results demonstrate that during infection, *B. cinerea* can secrete BcPLS1-positive EVs.

### *B. cinerea* secretes small RNA effectors through EVs

Our previous work demonstrated that *B. cinerea* can transfer sRNA effectors into host plant cells to suppress host immunity and promote infection[3]. To study whether *B. cinerea* uses EVs to transport these sRNA effectors, we examined fungal sRNAs using sRNA stem-loop reverse transcription-polymerase chain reaction (RT-PCR) analysis in EVs isolated from *B. cinerea* liquid culture[52]. Three previously discovered sRNA effectors, Bc-siR3.1, Bc-siR3.2, and Bc-siR5, were detected in *B. cinerea* EVs isolated by sucrose gradient fractionation. These sRNAs were concentrated primarily in fractions four and five, which also contained high levels of BcPLS1 (Fig. 2a). Additionally, EVs from fraction four were used for a Micrococcal Nuclease (MNase) protection assay. As shown in Fig. 2b, the sRNA effectors can still be detected after MNase treatment unless vesicles were ruptured with Triton X-100 (Fig. 2b and Supplementary Fig. 3). This result indicates that *B. cinerea* sRNAs are localized within and protected by EVs.

To assess the role of tetraspanin proteins in *B. cinerea* small RNA delivery, deletion mutants of *BcPLS1* and *BcTSP3* were generated by targeted gene replacement in wild-type *B. cinerea* strain B05 (Supplementary Fig. 4a). Two independent mutant strains of each gene Δ*bcpls1* and Δ*bctsp3* were complemented by expressing *BcPLS1* and *BcTSP3*, respectively (Supplementary Fig. 4a). On minimal medium (MM), but not potato dextrose agar, the growth rate of the Δ*bcpls1* strain was reduced compared to the wild-type and Δ*bctsp3* strains. (Supplementary Fig. 5). The spores of both mutants were collected and used to inoculate wild-type *Arabidopsis* plants to assess their virulence. The Δ*bcpls1* mutant strain showed a significant reduction in virulence with the lesion size being reduced by 84% compared with the wild-type strain (Fig. 2c). However, infection with Δ*bctsp3* resulted in lesions similar to the wild-type strain (Fig. 2c). To further understand the functions of

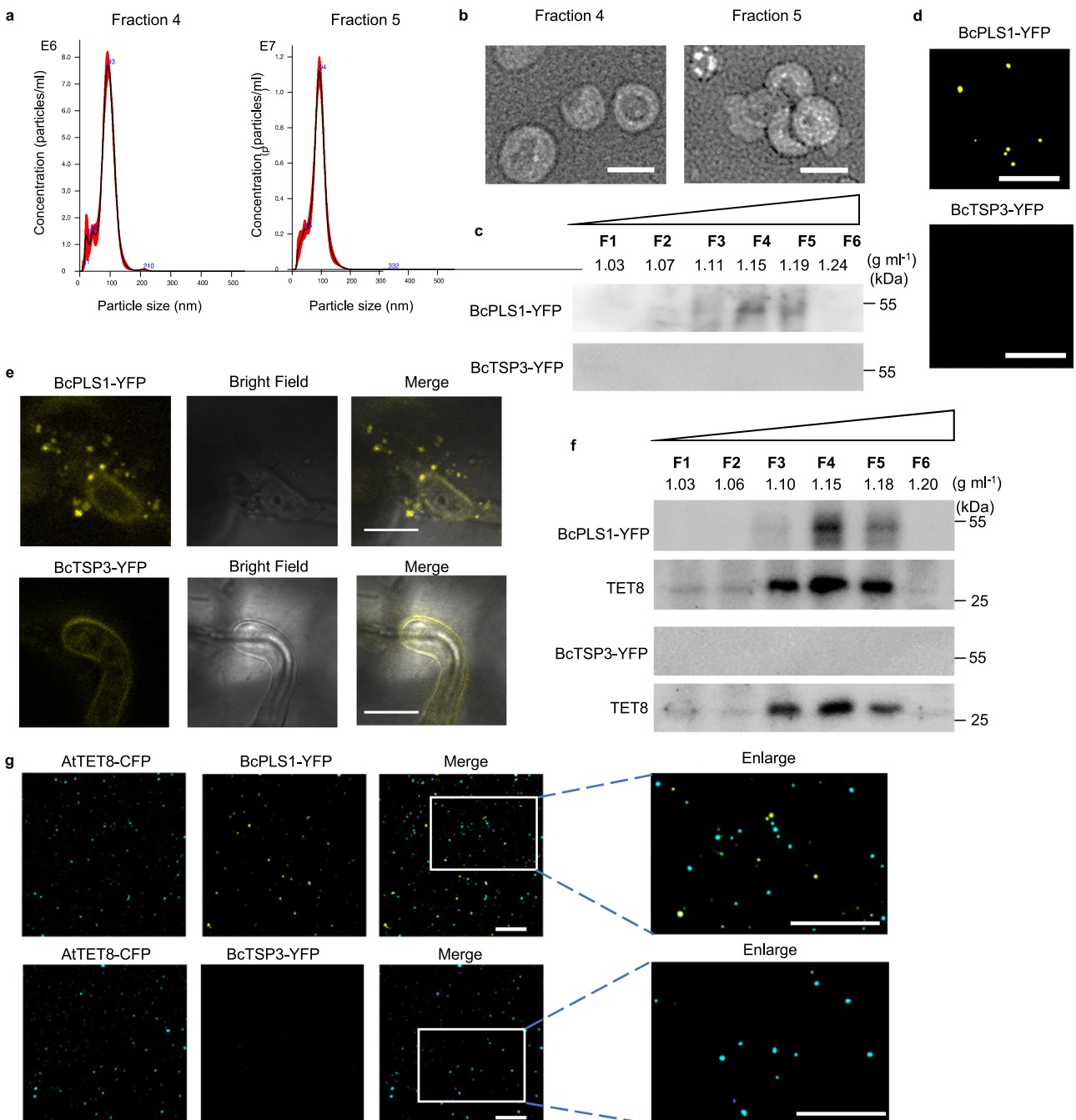

**Fig. 1 | *B. cinerea* secretes BcPLS1-positive EVs during infection. a** The size distributions of the *B. cinerea* EVs isolated by sucrose gradients in fractions 4 and 5 were measured using nanoparticle tracking analysis. The data of 4 measurements in one replicate are presented here. Similar data were obtained in three biological replicates. The red area represents the standard error. **b** Transmission electron microscopy images of *B. cinerea* EVs from fractions 4 and 5. Scale bar, 100 nm. **c** Western blot analysis of BcPLS1 and BcTSP3 in sucrose gradient fractions purified *B. cinerea* EVs from liquid culture. **d** *B. cinerea* EVs isolated from BcPLS1-YFP and BcTSP3-YFP liquid culture were examined by confocal microscopy. Scale bar, 10 μm. **e** BcPLS1-YFP labeled *B. cinerea* EVs were observed outside the fungal cells at the site of infection on *Arabidopsis* leaves. Samples were pretreated with 0.75 M sorbitol for 30 min to induce plasmolysis before imaging. Scale bars 10μm. **f** BcPLS1-YFP and BcTSP3-YFP were examined in sucrose gradient fractionated EVs that were isolated from BcPLS1-YFP or BcTSP3-YFP *B. cinerea* strains infected wild-type *Arabidopsis* using western blot. *Arabidopsis* TET8 was used as plant exosome control. **g** EVs isolated from BcPLS1-YFP or BcTSP3-YFP *B. cinerea* strains infected *Arabidopsis* TET8-CFP plants were purified using sucrose gradients, and the EVs from fraction four were examined using confocal microscopy. Scale bars, 10μm. Source data are provided as a Source Data file.

BcPLS1 and BcTSP3 in EV-mediated sRNA secretion from *B. cinerea*, EVs were isolated from the culture supernatants of both the *Δbcpls1* and *Δbctsp3* strains and Bc-sRNA amounts were examined by real-time PCR and sRNA stem-loop RT-PCR. Compared with the wild-type strain, significantly lower levels of Bc-sRNAs were detected in *Δbcpls1* EVs, while similar amounts of Bc-sRNAs were detected in *Δbctsp3* EVs (Fig. 2d and

Supplementary Fig. 6a). Further, we found the EV amount released by *Δbcpls1* was significantly reduced compared with wild-type and *Δbctsp3* strains (Supplementary Fig. 6b). This result indicates that BcPLS1 is important for EV-mediated sRNA secretion in *B. cinerea*.

To further confirm the function of BcPLS1 in *B. cinerea* virulence and EV secretion, we premixed *Δbcpls1* spores with EVs isolated from

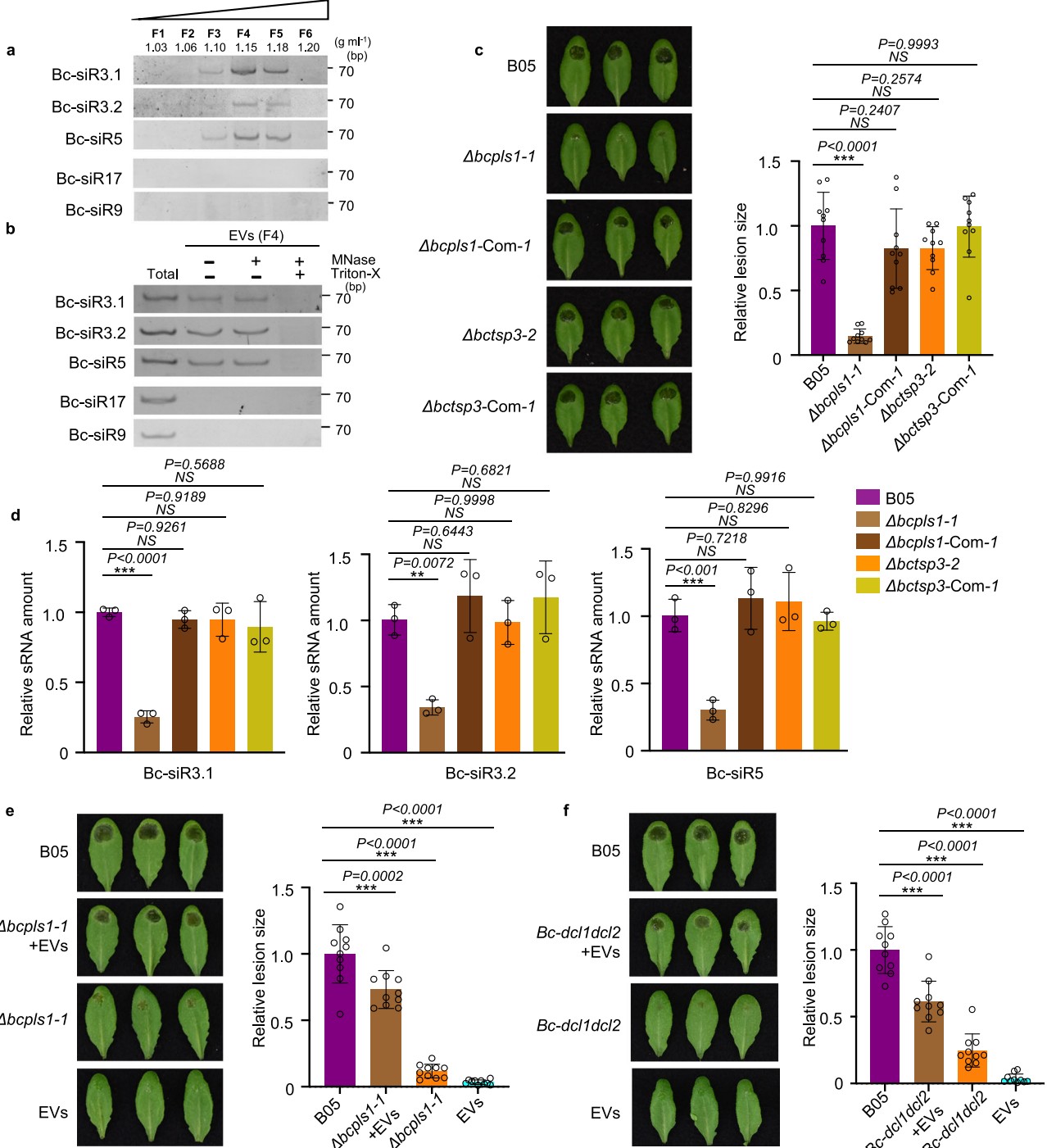

**Fig. 2 | _B. cinerea_ secretes small RNA effectors through EVs. a** _B. cinerea_ sRNAs were examined in _B. cinerea_ EVs isolated by sucrose gradient fractionation. **b** _B. cinerea_ sRNAs can be protected from micrococcal nuclease treatment by EVs. F4, _B. cinerea_ EV samples were collected from fraction 4 after sucrose gradient centrifugation. Bc-siR17 and Bc-siR9 were used as negative controls. **c** The _Δbcpls1_ strain showed strongly reduced virulence, whereas the _Δbctsp3_ strain showed similar virulence with the wild-type B05 strain. Both _Δbcpls1_ and _Δbctsp3_ complementary strains showed similar virulence with the wild-type B05 strain. **d** Bc-sRNAs were examined by Real-time PCR in EVs isolated from B05, _Δbcpls1_, _Δbctsp3_, _Δbcpls1_ complementary, and _Δbctsp3_ complementary strains. The data are presented as mean ± s.d. Similar results were obtained in three biologically independent experiments. **e** Wild-type _B. cinerea_ EVs partially complement the virulence of

_B. cinerea Δbcpls1_ mutant. Premixing of wild-type _B. cinerea_ EVs from sucrose gradient fraction 4 with _Δbcpls1_ mutant spores partially complement the weak virulence of _Δbcpls1_ on _Arabidopsis_ leaves. **f** Wild-type _B. cinerea_ EVs partially complement the virulence of _B. cinerea dcl1dcl2_ mutant. Premixing of wild-type _B. cinerea_ EVs from sucrose gradient fraction 4 with _dcl1dcl2_ mutant spores partially complements the weak virulence of _dcl1dcl2_ on _Arabidopsis_ leaves. Relative lesion size was determined 2 days after infection. The statistical data in **c**, **e** and **f** are presented as mean ± s.d., _n_ = 10 biological replicates. The statistical analysis was performed using ANOVA Dunnett's multiple comparisons test. The small open circles represent individual values. The error bars indicate s.d. **P < 0.01, ***P < 0.001. Source data are provided as a Source Data file.

wild-type *B. cinerea* and used the mixture to infect *Arabidopsis* plants. As shown in Fig. 2e, EVs from wild-type *B. cinerea* can partially restore the virulence of *Δbcpls1*. The lesion size increased from 12% of the wild-type strain to 73% of the wild-type strain. This result demonstrates that the cargoes in *B.cinerea* EVs are important for fungal virulence.

To determine whether sRNAs in the EVs contribute to the fungal virulence, we premixed spores from a *B. cinerea* mutant *dcl1dcl2* strain with EVs isolated from wild-type *B. cinerea* culture supernatants and then inoculated *Arabidopsis* plants. The *dcl1dcl2* mutant cannot generate sRNA effectors, such as Bc-siR3.1, Bc-siR3.2, and Bc-siR5, and exhibits a very weak virulence phenotype[3]. However, when premixed with EVs from wild-type *B. cinerea*, which contain sRNA effectors, *B. cinerea dcl1dcl2* mutant pathogenicity was dramatically recovered. The lesion size increased from 24% of the wild-type strain to 61% of the wild-type strain (Fig. 2f). We then further used the EVs isolated from the *dcl1dcl2* mutant to complement the phenotype of *Δbcpls1*. Results showed that *B. cinerea dcl1dcl2* mutant EVs could only restore the virulence of *Δbcpls1* from 18% of the wild-type strain to 32% of the wild-type strain (Supplementary Fig. 7). This modest increase may be attributed to the other cargoes in the *dcl1dcl2* EVs, because fungal EVs have been found to serve as a mode of transport for various virulence-related proteins and toxic metabolites[53]. These results indicate that fungal DCL1/DCL2-generated sRNA effectors encapsulated in *B. cinerea* EVs play an important role in fungal virulence.

## Clathrin-mediated endocytosis is involved in plant immunity against *B. cinerea*

In the mammalian system, multiple endocytic pathways are used to internalize EVs and their cargoes from various origins[54]. To determine whether endocytosis pathways are also involved in the uptake of *B. cinerea* EVs and their sRNA cargoes, we first treated wild-type *Arabidopsis* leaves with plant CME-specific inhibitor ES9-17 or CIE-specific inhibitor MβCD[55,56]. Vesicle internalization was strongly suppressed by ES9-17 but not by MβCD (Supplementary Fig. 8a, c), suggesting that CME is likely the major pathway for EV uptake. However, we found that ES9-17 can also suppress the growth of *B. cinerea* (Supplementary Fig. 8b). So, the chemical treatment may cause a broad impact on both plants and fungi. Therefore, we chose genetic methods to ascertain the involvement of the CME pathway in the plant immune response to *B. cinerea* infection. We performed *B. cinerea* infection on two *Arabidopsis* CME pathway mutants, *chc2* and *ap2σ*, which carry mutations in *CLATHRIN HEAVY CHAIN 2* and *ADAPTOR 2 SIGMA SUBUNIT*, respectively. Compared with wild-type plants, both *chc2-1* and *ap2σ* mutants displayed reduced disease susceptibility to *B. cinerea* (Fig. 3a). The relative lesion size on *chc2-1* and *ap2σ* mutants was significantly reduced by 60% and 73% compared with wild-type plants, respectively (Fig. 3a). Furthermore, we took advantage of inducible lines that expressed the C-terminal part of clathrin heavy chain (INTAM≫RFP-HUB1) or the clathrin coat disassembly factor Auxilin2 (AX2) (XVE≫AX2). Overexpression of both lines blocked CME in *Arabidopsis*[41,57]. After induction, both INTAM≫RFP-HUB1 and XVE≫AX2 lines showed significantly reduced susceptibility to *B. cinerea* infection (Fig. 3b, c). The relative lesion size on INTAM≫RFP-HUB1 and XVE≫AX2 was reduced by 45% and 59% compared with control plants. These results confirm that CME contributes to plant susceptibility to *B. cinerea* infection.

Next, we tested the possible function of the CIE pathway in plant immunity against *B. cinerea* infection. We examined the response of CIE-pathway mutants *flot1/2*, *remorin 1.2*, and *remorin 1.3* after infection by wild-type *B. cinerea*. Because there is a close homolog of *FLOT1* in *Arabidopsis* named *FLOT2*, we generated *flot1/2* double mutants by knocking out *FLOT2* in the *flot1* mutant background using the CrisprCas9 system (Supplementary Fig. 9). Two independent lines were inoculated by *B. cinerea*, both of them showed a slightly more susceptible phenotype compared with wild-type (Fig. 3d). The homozygous knockout mutants of *rem1.2* and *rem1.3* T-DNA insertion lines

were available and used for pathogen assays. No noticeable difference was observed between *rem1.2* or *rem1.3* and wild-type plants when challenged by *B. cinerea* spores (Fig. 3e). These results indicate that CME plays a more important role than CIE in plant susceptibility and the internalization of *B. cinerea* sRNAs into plant cells.

## Bc-sRNA loading into host AGO1 and subsequent host target gene suppression is attenuated in CME mutants

*B. cinerea* sRNAs can hijack the host plant RNAi machinery and silence plant immunity-related genes by binding to *Arabidopsis* Argonaute protein AGO1[3]. If CME is a critical factor in the internalization of EVs and Bc-sRNAs into plant cells, the deficiency of CME is likely to impact the level of sRNAs incorporated into the AGO1 protein. To determine whether CME is crucial for the uptake of *B. cinerea* sRNAs, we immunoprecipitated AGO1 from *B. cinerea*-infected wild-type *Arabidopsis* and CME mutants, including *chc2-1* and *ap2σ*. As shown in Fig. 4a, the fungal sRNA effectors, Bc-siR3.1, Bc-siR3.2, and Bc-siR5, were all detected in the AGO1-associated fraction precipitated from infected wild-type plants. However, the amount of *B. cinerea* sRNAs loaded into AGO1 was significantly reduced by around 60% and 80% in *chc2-1* and *ap2σ*, respectively, compared to wild-type plants. This observation was further supported by similar results obtained from INTAM≫RFP-HUB1 and XVE≫AX2 lines (Fig. 4b, c).

We also assessed whether CIE contributes to Bc-sRNA uptake by challenging *flot1/2*, *rem1.2*, and *rem1.3* mutants with wild-type *B.cinerea*. Unlike CME mutants, the amount of AGO1-associated Bc-sRNAs in *flot1/2*, *rem1.2*, and *rem1.3* mutants was similar to wild-type plants (Fig. 4d, e). As the total amount of AGO1 protein was not decreased in any of the CME and CIE mutants (Supplementary Fig. 10), the decrease of Bc-sRNAs associated with AGO1 in CME mutants can be attributed to a reduction in the Bc-sRNA internalization of plant cells, rather than a decrease in the amount of AGO1 protein. Thus, CME is a key pathway for Bc-sRNA trafficking into host plant cells.

*B. cinerea* sRNA effectors have a suppressive effect on predicted host target genes during infection[3,58]. If CME is truly essential for cross-kingdom trafficking of Bc-sRNAs, we expect to observe attenuated suppression of host target genes in CME mutants. Thus, we quantified the transcript levels of Bc-siR3.1 target gene *At-PRXIIF*, Bc-siR3.2 target genes *At-MPK1* and *At-MPK2*, and Bc-siR5 target gene *At-WAK*, at 0 and 24 hours post-infection. As expected, the expression of these genes was suppressed in wild-type plants after 24 hours of infection. However, this suppressive effect was attenuated in the *chc2-1*, *ap2σ*, induced INTAM≫RFP-HUB1 and XVE≫AX2 mutants shown by no reduction in target gene transcript levels 24 hours after infection (Fig. 4f and Supplementary Fig. 11a, b). We performed a similar experiment to explore the role of the CIE pathway in target gene suppression using the *flot1/2*, *rem1.2*, and *rem1.3* mutants. Unlike CME mutants, all target genes were still significantly suppressed in the CIE mutants (Supplementary Fig. 12a, b). These results again demonstrate that CME is the major pathway for Bc-sRNA uptake.

## *B. cinerea* induces accumulation of clathrin-coated vesicles around infection sites

To further validate the role of CME in the internalization of *B. cinerea* sRNA effectors into host plant cells, *Arabidopsis CLATHRIN LIGHT CHAIN 1* (*CLC1*)-GFP and *CLATHRIN HEAVY CHAIN 2* (*CHC2*)-YFP transgenic plants were challenged by *B. cinerea*. Eight hours post inoculation, the infected plants were stained with the endocytic tracer FM4-64 and subjected to confocal microscopy to examine the role of endocytosis in plant response[38]. Numerous distinct fluorescently labeled endocytic vesicles were observed around *B. cinerea* infection sites, indicating the activation of endocytosis during *B. cinerea* infection (Fig. 5a, b). Meanwhile, we detected co-localization between CLC1-GFP or CHC2-YFP with the FM4-64 signal on vesicles at the infection sites, with co-localization rates of 52% and 60%, respectively (Fig. 5a, b).

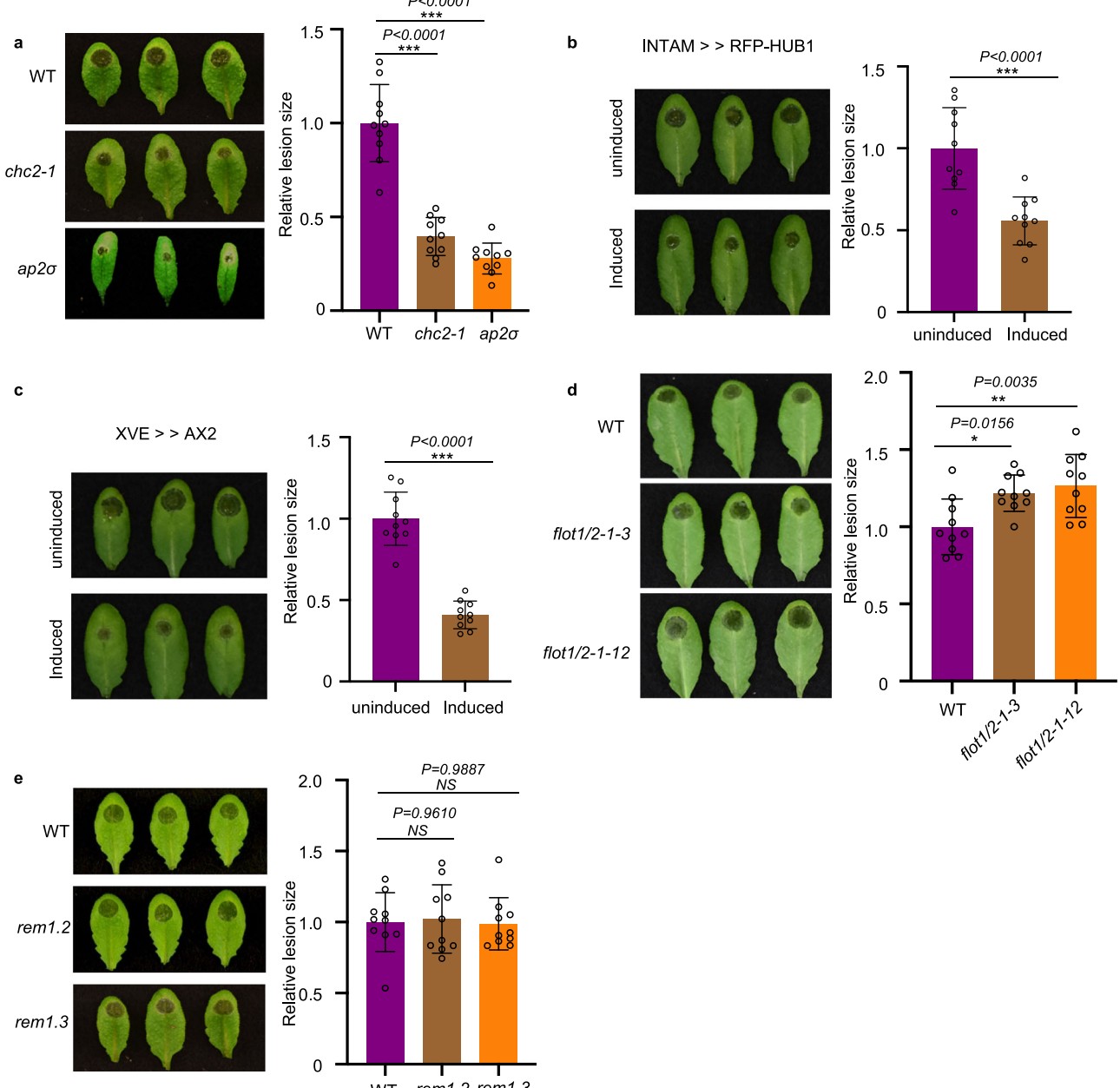

**Fig. 3 | Clathrin-mediated endocytosis (CME) is involved in plant susceptibility against *B. cinerea*. a** *Arabidopsis chc2-1* and *ap2σ* mutants exhibited enhanced disease resistance to *B. cinerea* infection. **b, c** INTAM≫RFP-HUB1(**b**) and XVE≫AX2 (**c**) transgenic plants showed enhanced resistance phenotype to *B. cinerea* infection after induction by 2 μM 4-Hydroxytamoxifen or 5 μM estradiol, respectively. **d** *Arabidopsis flot1/2* double mutants showed increased susceptibility to *B. cinerea* infection. **e** *Arabidopsis rem1.2* and *rem1.3* mutants exhibited similar susceptibility to *B. cinerea* compared with the wild-type. Relative lesion size was determined by comparing lesion size to the size of the entire leaf 2 days after infection. The data are presented as mean ± s.d., *n* = 10 biological replicates. The statistical analysis in **b** and **c** was performed using unpaired two-tailed Student's t-test. The statistical analysis in **a**, **d** and **e** were performed using ANOVA Dunnett's multiple comparisons test. The small open circles represent individual values. The error bars indicate s.d. \**P* < 0.05, \*\**P* < 0.01, \*\*\**P* < 0.001. Source data are provided as a Source Data file.

These results suggest that *B. cinerea* infection induces the accumulation of CCVs around infection sites. In addition to clathrin, AP2 adaptor complexes are important components for CME in plant cells[59]. AP2A1-GFP plants were also treated with wild-type *B. cinerea* spores. Similar to CHC2-YFP and CLC1-GFP plants, clear colocalization (46%) was observed on the endocytic vesicles at the infection sites between AP2A1-GFP with FM4-64 (Fig. 5c). We also generated the YFP-tagged FLOT1 transgenic line and monitored the endocytosis through FM4-64 staining following inoculation with *B. cinerea*. As shown in Fig. 5d, endocytic FM4-64 signals can also be observed, but the percentage of colocalization (10%) between YFP-FLOT and FM4-64 was limited. These

results further confirm that CME is more involved than CIE in *B. cinerea* infection of *Arabidopsis*.

## *B. cinerea* EVs and sRNAs are associated with clathrin-coated vesicles

Unlike in animal cells, plant CCVs are not disassembled immediately after scission from the plasma membrane, which allows the isolation of CCVs by immunocapture[60]. We isolated plant CCVs from CLC1-GFP transgenic plants after infection with *B. cinerea* using a two-step procedure comprising differential centrifugation, followed by immunoaffinity purification with antibodies against GFP. GFP signal was

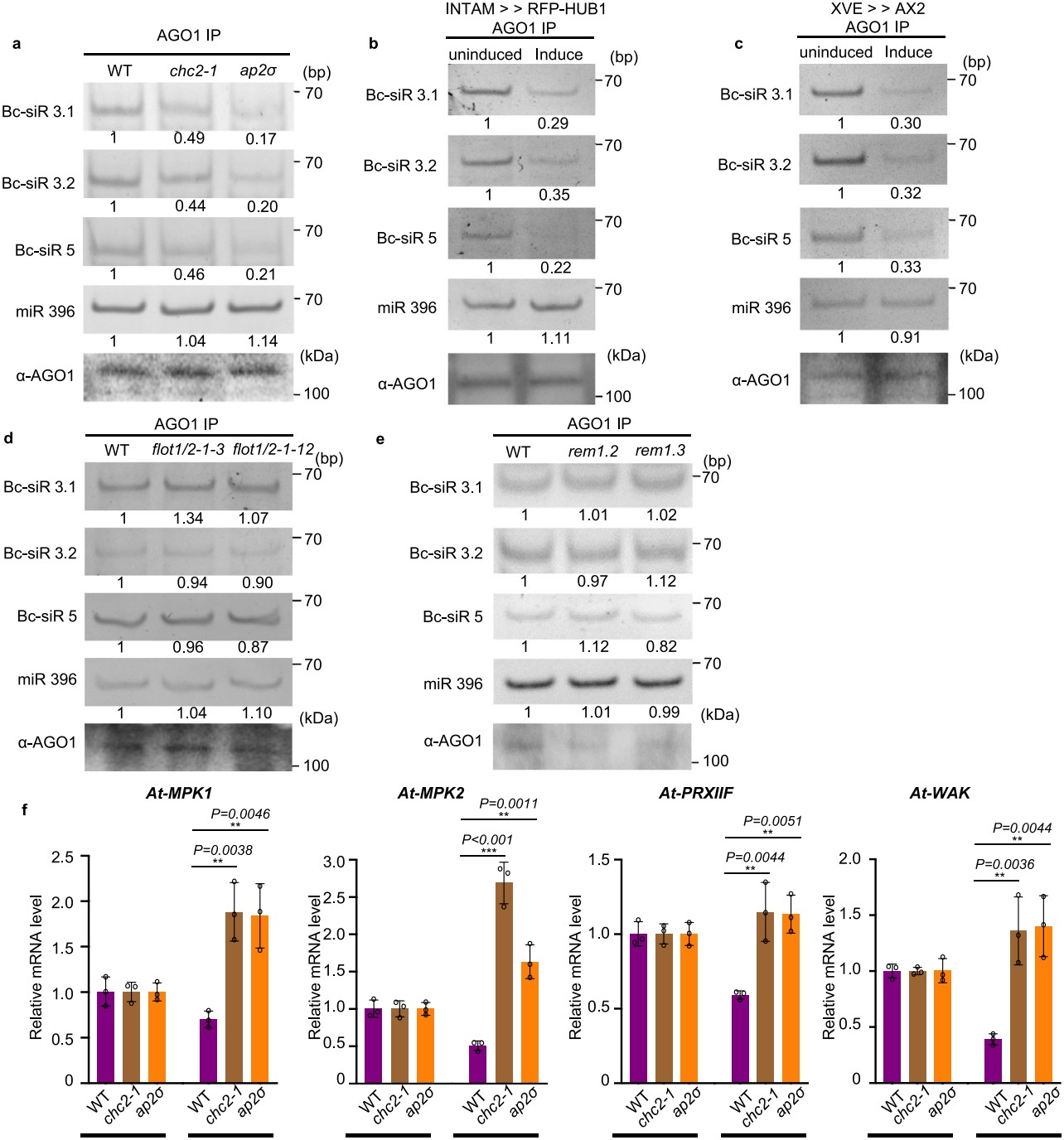

**Fig. 4 | Bc-sRNA loading into plant AGO1 and target gene suppression was reduced in CME mutants. a** The loading of Bc-siR3.1, Bc-siR3.2, and Bc-siR5 into AGO1 of wild-type, *chc2-1*, and *ap2σ* mutants was examined with AGO1-IP followed by RT-PCR. **b, c,** The loading of Bc-siR3.1, Bc-siR3.2, and Bc-siR5 into AGO1 of induced INTAM≫RFP-HUB1 (**b**) and XVE≫AX2 (**c**) transgenic plants was examined with AGO1-IP followed by RT-PCR. **d, e** The loading of Bc-siR3.1, Bc-siR3.2, and Bc-siR5 into AGO1 of wild-type, *flot1/2* (**d**), *rem1.2*, and *rem1.3* mutants (**e**) was examined with AGO1-IP followed by RT-PCR. In **a–e**, miR396 served as a positive control. AGO1 was detected by western blot. **f**, Suppression of host target genes by Bc-sRNAs was attenuated in CME mutants. The expression of Bc-siR3.1 target *At-PRXIIF*, Bc-siR3.2 target *At-MPK1* and *At-MPK2*, and Bc-siR5 target *At-WAK* in *chc2-1* and *ap2σ* mutants compared with those in wild-type infected plants after *B. cinerea* infection was examined by real-time RT-PCR. The data are presented as mean ± s.d. Similar results were obtained in three biologically independent experiments. The statistical analysis was performed using ANOVA Dunnett's multiple comparisons test. The small open circles represent individual values. The error bars indicate s.d. *P< 0.05, **P< 0.01, ***P< 0.001. Source data are provided as a Source Data file.

clearly detected on the surface of GFP-TRAP agarose beads after incubation with isolated CCVs (Fig. 6a). To examine whether the immuno-captured CCVs were intact, the GFP-TRAP beads were subjected to scanning electron microscopy (SEM) analysis after pull down. Figure 6b shows that most of the immuno-captured CCVs were intact

with typical cage structures on the surface of GFP-TRAP agarose beads. We further inoculated CLC1-GFP plants with the BcPLS1-mCherry *B. cinerea* strain. After ten hours of inoculation, colocalization of CLC1-GFP and BcPLS1-mCherry signals could be observed around the infection sites (Fig. 6c). Further, BcPLS1-mCherry can be detected in

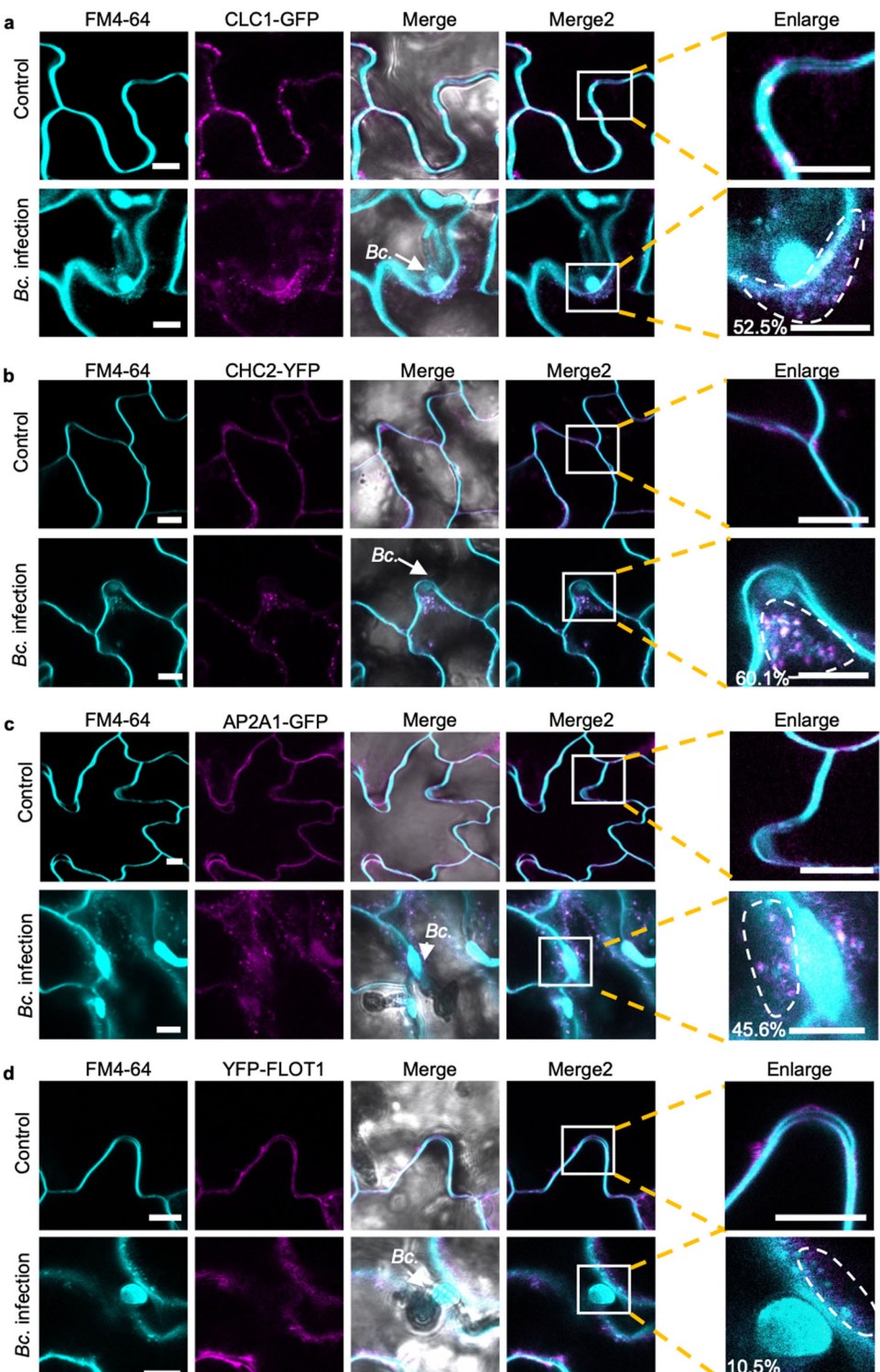

**Fig. 5 | Clathrin-coated vesicles accumulate around *B. cinerea* infection sites.**
**a–c** Confocal microscopy showed that the CLC1-GFP (**a**), CHC2-YFP (**b**) and AP2A1-GFP (**c**) signal overlapped with endocytic FM4-64 vesicle signals when treated with *B. cinerea*. Untreated plants served as control. **d** Confocal microscopy of *B. cinerea* infection on *Arabidopsis* YFP-FLOT1 plants. The scale bars in **a**–**d**, are 10 μm. The arrows indicate the location of *B. cinerea* (Bc.). The degree of colocalization between CLC1-GFP, CHC2-YFP, AP2A1-GFP, and YFP-FLOT1 and FM4-64 signals was quantified using three independent images. Regions of interest (ROIs) that are used for measuring colocalization percentage were marked in the enlarged panel. Source data are provided as a Source Data file.

the CCV fractions isolated from infected plants (Fig. 6d). The Bc-sRNA effectors that we examined previously were also present in the immunopurified CCVs (Fig. 6e). These results provide conclusive evidence that *B. cinerea* EVs are responsible for transporting Bc-sRNAs and enabling their internalization through plant CME.

## Discussion

Cross-kingdom RNAi is emerging as a critical regulatory mechanism across interaction systems. While recent studies show that plant and animal hosts utilize extracellular vesicles to protect and transport sRNAs to their fungal pathogens[2,5], how fungi transported sRNAs to

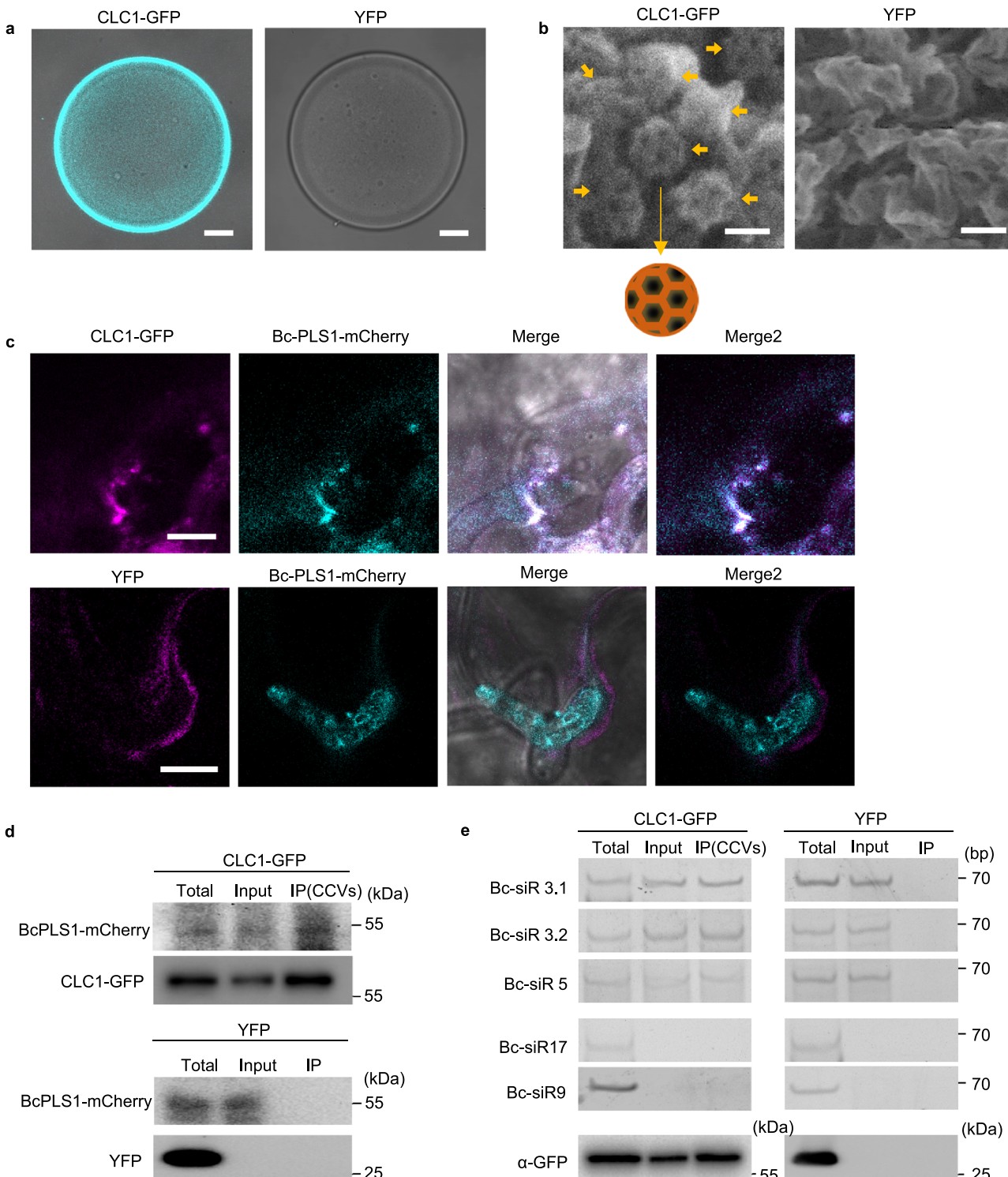

**Fig. 6 | CME mediates the uptake of _B. cinerea_ secreted sRNAs. a** CLC1-GFP labeled CCVs can be pulled down by GFP-Trap Agarose beads. _Arabidopsis_ YFP plant was used as a negative control. Scale bar, 10 μm. **b** Intact CCVs can be observed on the surface of GFP-Trap Agarose beads after CCV pull down from CLC1-GFP plants using the scanning electron microscope (SEM). A schematic figure of CCV is presented. The solid arrows indicate different intact CCVs. Scale bar, 100 nm. **c** BcPLS1-mCherry colocalizes with _Arabidopsis_ CLC1-GFP during infection. **d** BcPLS1-mCherry signals can be detected in GFP-Trap Agarose beads-isolated CCVs from BcPLS1-mCherry infected CLC1-GFP plants using western blot. YFP plant was used as a negative control. **e** _B. cinerea_-secreted sRNAs can be detected in purified plant CCVs by RT-PCR. In **d** and **e,** the crude CCVs fraction obtained from _B. cinerea_-infected CLC-GFP or YFP plants through sucrose gradients was used for immunoprecipitation. Bc-siR17 and Bc-siR9 were negative controls. CLC1-GFP and YFP protein was detected by western blot using an anti-GFP antibody. Source data are provided as a Source Data file.

their hosts was still unknown. Here, we demonstrate that, just like their plant hosts, *B. cinerea* utilizes EVs to protect and transport sRNAs to plants, providing the first example of fungi utilizing EVs for cross-kingdom RNA transport. Recently, the algal species *Emiliania huxleyi* was found to secrete vesicles containing sRNAs that influence viral infection and the population dynamics of interacting marine microorganisms[61,62]. Furthermore, bacterial EVs isolated from seawater were found to transport DNA transposons that facilitate horizontal gene transfer of nutrient-acquisition and bacteriophage-resistance genes among subpopulations of marine picocyanobacterium *Prochlorococcus*[63]. These studies demonstrate that EVs can survive in harsh environment and provide perfect protection for vulnerable RNA and DNA during transport, highlighting the under-appreciated role of EVs in cross-kingdom/organismal communication, host-microbial interaction and microbial evolution[23].

We show that plant cells internalize fungal EVs through clathrin-mediated endocytosis. CME is one of the major pathways for plant uptake of environmental molecules, with extensive studies demonstrating its vital role in different biological processes[38,64], including plant defense responses[39,41,65–67]. While the plant CME pathway has recently been implicated in the uptake of protein effectors from fungal and oomycete pathogens[44,45], we demonstrate here that it is also critical for the plant uptake of fungal sRNA effectors and fungal EVs. Although in mammals, intraorganismal EV internalization into recipient cells can occur through CME[68], the mechanisms of internalization of foreign EVs from one organism to another in a different kingdom had never been reported until this study. We found that numerous CCVs accumulated around infection sites, indicating the activation of CME during *B. cinerea* infection. Additionally, the *B. cinerea* EV marker protein BcPLS1 is colocalized with plant CME component protein CLC1 during fungal infection and can be detected in the immunopurified plant CCVs. These findings strongly support that fungal EVs are internalized by the plant CME pathway. Furthermore, the EV-delivered sRNAs were also detected in the immunopurified CCVs, providing additional evidence for the role of plant CME in the uptake of fungal EV-delivered sRNAs.

These new insights into the mechanistic basis for cross-kingdom RNA communication between fungi and plants can help in the development of translational applications. For example, in *B. cinerea* and many other pathogens, such as *Fusarium graminearum, Penicillium italicum, Colletotrichum gloeosporioides*, and *Valsa mali*, sRNA effectors are generated by DCL proteins and used to promote virulence, and mutation in the *DCL* genes attenuates the virulence of these pathogens[3,69–72]. Consequently, targeting and silencing *DCL* and other components of the RNAi machinery, even transiently, has been proven to provide powerful protection against these pathogens by hampering their ability to synthesize sRNA effectors[16,18–22]. Specifically, using spray-induced gene silencing (SIGS) to suppress the expression of both fungal *DCL1* and *DCL2* genes can effectively protect plants from infection by various pathogens, including fungal pathogens *B. cinerea*[16,18–20], *F. graminearum*[21], and oomycete pathogen *Plasmopara viticola*[22]. Recent application of modern nanotechnology in SIGS[20,73–75], especially the utilization of artificial nanovesicles that mimic EVs for RNA protection[19,73], largely improves the efficacy of SIGS for crop protection. Here, by elucidating how sRNA effectors are transported from fungal pathogens into plants, we provide additional mechanistic targets for disease control. Specifically, we demonstrate that the EV-associated *B. cinerea* tetraspanin protein, BcPLS1, is an important protein for fungal EV secretion and virulence. Previous studies on PLS1 in both the *B. cinerea* T4 strain and *Magnaporthe grisea* showed that *Δpls1* mutants exhibit weak virulence[76,77], and the *M. grisea* PLS1 protein was found to be mainly expressed during infection and localized to appressorial membranes[77]. Consistently, we found the *Δbcpls1* mutant showed weak virulence on plants (Fig. 2c) and the BcPLS1 protein can be observed on the appressorial membranes (Fig. 1e). Furthermore, we detected BcPLS1 in the isolated *B. cinerea* EVs, and found that the

delivery of *B. cinerea* sRNA effectors is compromised in the *Δbcpls1* mutant, suggesting that BcPLS1-positive EVs are critical for sRNA effector transport. Thus, we have expanded the repetoire of effective genetic targets, such as PLS1, for RNAi-based anti-fungal strategies such as SIGS. Combining existing effective gene targets with PLS1 and other EV proteins targeting in SIGS will likely provide even more effective crop protection by targeting not only sRNA biogenesis, but also the trafficking mechanism of sRNA effectors from pathogens to plants.

Overall, our study helps to elucidate the underlying molecular mechanisms of EV-mediated sRNA trafficking during "cross-kingdom RNAi" and will further inform efforts in other organisms that also deliver sRNAs into interacting species. Given how widespread the phenomenon of cross-kingdom RNAi is in different branches of life, this will have strong rammifications for anti-pathogen measures in agriculture and human health.

## Methods

### Fungal strains and culture conditions

*Botrytis cinerea* strain B05.10 was used as a host strain for gene replacement and fluorescent labeling experiments, and as a wild-type (WT) control in all experiments. WT, mutants, and transgenic strains were all grown on complete media Potato Dextrose Broth with 1.5% Agar (PDA) at room temperature. The minimal medium (MM) utilized for measuring growth rate was prepared according to a previous study[78]. The MM was prepared as follows (per liter of distilled $H_2O$): sucrose, 30 g; $KH_2PO_4$, 1 g; $MgSO_4 \cdot 7H_2O$, 0.5 g; KCl, 0.5 g; $FeSO_4 \cdot 7H_2O$, 10 mg; $NaNO_3$, 2 g; agar, 20 g; trace element solution, 0.2 ml. The trace element solution contains (per 95 ml of distilled $H_2O$): citric acid, 5 g; $ZnSO_4 \cdot 7H_2O$, 5 g; $Fe(NH_4)_2(SO_4)_2 \cdot 6H_2O$, 1 g; $CuSO_4 \cdot 5H_2O$, 0.25 g; $MnSO_4 \cdot H_2O$, 50 mg; $H_3BO_4$, 50 mg; $NaMoO_4 \cdot 2H_2O$, 50 mg.

### EV isolation

EV isolation from fungal culture supernatant was performed as previously described[27]. In order to eliminate the possibility of contamination from the culture medium, the medium was initially ultracentrifuged at 100,000 g for 1 hour to eliminate any particles. Subsequently, the medium was filtered using a 0.22 μm filter prior to its use for *B. cinerea* culture. $10^6$ *B. cinerea* spores were cultured for 48 hours at room temperature with shaking on the rocker in 100 ml of purified YEPD (0.3% yeast extract, 1% peptone, 2% glucose, pH 6.5) medium. The culture supernatant was collected by centrifugation at 3000 g for 15 mins at 4 °C. The supernatant was then filtered through a 70 μm cell strainer to further remove cells. The cell-free supernatant was centrifuged at 10,000 g for 30 mins to remove smaller debris and a second round of centrifugation at 10,000 g for 30 mins was performed to ensure the removal of aggregates. The resulting supernatant was filtered through a 0.45 μm filter and ultracentrifuged at 100,000 g for 1 h to precipitate vesicles. Vesicles were washed once in phosphate-buffered saline (PBS, 137 mM NaCl, 2.7 mM KCl, 10 mM $Na_2HPO_4$, 1.8 mM $KH_2PO_4$), and the final pellets were resuspended in PBS.

EV isolation from infected plants was performed as previously described[2,49]. Apoplastic fluid was extracted from *B. cinerea*-infected *Arabidopsis* leaves by vacuum infiltration with infiltration buffer (20 mM MES, 2 mM $CaCl_2$, 0.1 M NaCl, pH 6.0) and then centrifuged at 900 g for 10 mins. Cellular debris was further removed from the supernatant by centrifugation at 2000 g for 30 minutes, filtration through a 0.45 μm filter and finally centrifugation at 10,000 g for 30 mins. EVs were then collected by ultracentrifugation at 100,000 g for 1 hr and washed with infiltration buffer at 100,000 g for 1 hour.

### Sucrose gradient separation of EVs

EVs from *B. cinerea* liquid culture and *B. cinerea*-infected *Arabidopsis* were purified by discontinuous sucrose density gradient centrifugation[25]. 10–90% sucrose stocks (w/v), including 10, 16, 22, 28,

34, 40,46, 52, 58, 64, 70 and 90%, were prepared using infiltration buffer (20 mM MES, 2 mM CaCl2, 0.1 M NaCl, pH 6.0). The discontinuous gradient was prepared by layering 1 ml of each solution in a 15-ml ultracentrifuge tube. 100 μl EVs were premixed with 1 ml of 10% sucrose stock and loaded on top of the sucrose gradient. Then, samples were centrifuged in a swinging-bucket rotor for 16 h at 100,000 *g*, 4 °C and six fractions (2 ml each) were collected from top to bottom. Collected fractions were transferred to new ultracentrifuge tubes and each sample was diluted to 12 ml using infiltration buffer, followed by a final centrifugation for 1 h at 100,000 g, 4 °C to obtain pellets. The pellet was then resuspended in 50 μl infiltration buffer for further analysis.

### Nanoparticle tracking analysis

A NanoSight NS300 fitted with a blue laser (405 nm) with NanoSight NTA software version 3.1 (Malvern Panalytical) was used to determine the size distribution and concentration of the *B. cinerea* EVs. EV samples were diluted 50 times using 0.22 μm filtered PBS and injected into the flow cell at a flow rate of 50 units. Four 60 second videos were taken for each sample and used to determine the size distribution and concentration of the EVs.

### Transmission electron microscopy analysis

*B. cinerea* EVs were imaged using negative staining. Briefly, 10 μl of EVs were deposited onto 3.0 mm copper Formvar-carbon-coated grids (EMS) for 1 min and excess sample was removed from the grids using filter paper. The grids were then stained with 1% uranyl acetate for 30 s and blotted before a second round of staining with 1% uranyl acetate 2 min. The samples were air-dried before visualization with a Talos L120 transmission electron microscope, operated at 120 kV.

### sRNA expression analyses

RNA was extracted using the Trizol extraction method for both total RNA and EV samples. sRNA RT-PCR was performed as previously described[2]. The PCR products were detected on a 12% PAGE gel. Quantitative PCR was performed with the CFX96 real-time PCR detection system (Bio-Rad) using SYBR Green mix (Bio-Rad). All primer sequences are provided in Supplementary Data 1.

To determine if sRNAs were localized within vesicles, EVs from sucrose gradient-purified fraction four were treated with 10 U of micrococcal nuclease (MNase) (Thermo Fisher) with or without Triton-X-100. For Triton-X-100 treatment, vesicles were incubated with 1% Triton-X-100 on ice for 30 minutes before nuclease treatments. Nuclease treatment was carried out at 37 °C for 15 minutes followed by RNA isolation.

### Transformation

The homologous recombination-based method was used to knock out *B. cinerea* genes as described previously[3]. The 5′- and 3′-regions of *BcPLS1* and *BcTSP3* were amplified and inserted into a modified pBluescript KS(+) plasmid (PBS-HPH) in which a hygromycin resistance cassette containing the *hph* gene of *E. coli* under the control of the *trpC* promoter of *Aspergillus nidulans* had been inserted between HindIII and SalI digestion sites. The full-length fragments containing the 5′- and 3′-regions of each specific gene and the *hph* cassette were transformed into the protoplasts of wild-type *B. cinerea*. The transformants were purified by the isolation of a single spore on PDA containing 70 μg ml⁻¹ hygromycin.

The complementary lines were generated by expressing the CDS of BcPLS1 or BcTSP3 in the Δ*bcpls1* or Δ*bctsp3* mutants, respectively. The CDS of both BcPLS1 and BcTSP3 were cloned separately into the pENTR vector (Life Technologies), then into the destination vector pBCC, which carries nourseothricin (NTC) N-acetyl transferase (*NAT*) conferring nourseothricin resistance. The specific primers 4-5-S and 4-3-A (Supplementary Data 1) were used to amplify dsDNA fragments

containing promotor *PtrpC*, inserted genes, and *NAT* for *B. cinerea* transformation The positive transformants were selected on PDA plates with 50 μg ml⁻¹ Nourseothricin Sulfate (Goldbio) and 70 μg ml⁻¹ hygromycin.

The BcPLS1-YFP and BcTSP3-YFP strains were generated by amplifying and inserting the stop codon removed CDS region and 3′ UTR region into the modified PBS-HPH, in which the YFP gene sequence has been inserted between BamHI and HindIII. The BcPLS1-mCherry was generated by inserting the stop codon removed CDS region and 3′ UTR region into the modified PBS-HPH with the mCherry sequence inserted at the C terminal. The transformation and selection conditions were the same as the homologous recombination-based gene knock-out. All primers used are provided in Supplementary Data 1.

### Plant materials and growth conditions

*Arabidopsis thaliana* wild-type and mutant lines used in this study were all Columbia 0 (*Col-O*) backgrounds. The *chc2-1*(SALK_028826)[79], *ap2σ* (SALK_141555)[59], *rem1.2* (SALK_117639)[80], *rem1.3* (SALK_023886C)[80] T-DNA insertion lines were ordered from the Arabidopsis Biological Resource Center (Ohio State University, USA). The CLC1-GFP[81] and AP2A1-GFP[81] lines were provided by Dr. Eugenia Russinova's lab. The INTAM > > RFP-HUB1[57] and XVE > > AX2[41] lines were provided by Dr. Jiří Friml. The CHC2-YFP, TET8-CFP, and YFP-only transgenic lines were constructed using the *Agrobacterium*-mediated flower dipping method as described previously[82]. The *flot1/2* double mutant was generated using Crispr-Ca9 system in the *flot1* (SALK_203966C) mutant background. The designed guide RNA sequences are listed in Supplementary Data 1. Both wild-type and mutant plants were grown at 22 °C under 12 h light per day.

### Plasmid construction

The vector for CHC2-YFP was constructed as follows: *Arabidopsis* genomic DNA was used as a template with primers CHC2-Pentr-F and CHC2-Pentr-R to amplify the whole sequence of CHC2. The final PCR product was cloned into pENTR/SD/D-TOPO (Life Technologies), and then into the destination vector pEarleyGate 101 by LR reactions (Life Technologies). For the YFP-FLOT1 construct, the full-length CDS of FLOT1 was cloned into the pENTR/SD/D-TOPO vector (Life Technologies), then into the destination vector pEARLYGATE 104 for YFP tagging (Life Technologies). For the TET8-CFP construct, the full-length CDS of TET8 was cloned into the pENTR/SD/D-TOPO vector (Life Technologies), then into the destination vector pEARLYGATE 102 for CFP tagging (Life Technologies). For the construct to express YFP, the YFP sequence was amplified from pEARLYGATE 101 and then cloned into pENTR/SD/D-TOPO vector and finally into pEARLYGATE 100 (Life Technologies). Primer sequences are listed in Supplementary Data 1.

### Chemical solutions and treatments

ES9-17 (Sigma Aldrich) was dissolved in DMSO to create a 30 mM stock solution. mβCD (Sigma Aldrich) was dissolved in deionized water to create a 200 mM stock solution. 4-Hydroxytamoxifen (Sigma Aldrich) was dissolved in DMSO to make a 2 mM stock solution. Estradiol (Sigma Aldrich) was dissolved in DMSO to make a 5 mM stock solution. For the chemical inhibitor treatments, *Arabidopsis* plant leaves were incubated in liquid 1/2 Murashige and Skoog (MS) medium with 30 μM or 10 mM mβCD for 24 hours. DMSO and deionized water were used as controls. To induce the expression of *RFP-HUB1* or *AX2*, 2 μM 4-Hydroxytamoxifen or 5 μM estradiol was used to pretreat the plant leaves in liquid 1/2 MS medium for 24 hours.

### CCV purification

The clathrin-coated vesicle isolation procedure was adapted from previously established protocols[83,84]. 50 grams of *Arabidopsis* leaves were homogenized for 1 min in 2 volumes of resuspension buffer (100 mM

MES, pH 6.4, 3 mM EDTA, 0.5 mM MgCl2, 1 mM EGTA, protease inhibitors (Roche)) using a Waring Blender. The homogenate was squeezed through eight layers of cheesecloth and three layers of Miracloth (Sigma). Then the homogenate was centrifuged in four increments to obtain a microsomal pellet: 500 g, 5 min; 5000 g, 10 min; 20,000 g, 30 min; 100,000 g, 60 min. The pellet was then resuspended in 6 ml resuspension buffer. Then the resuspended pellet was loaded on top of the discontinuous sucrose gradients, which were made from three sucrose stocks (10%, 35%, and 50%) in the resuspension buffer. The gradient was composed of 1 ml of 50% sucrose, 5 ml of 35% sucrose, and 3 ml of 10% sucrose in a Beckman 14 × 95 mm Ultra-Clear centrifuge tube. The gradients were centrifuged at 116,000 g 50 min. Samples were collected from 10% and 10/35% interface and resuspended in the resuspension buffer to make the sucrose concentration lower than 5%. The samples were then centrifuged at 200,000 g for 50 min, and the pellets were collected as crude CCVs.

Further immunoisolation of CCVs was performed as follows: GFP-Trap® Agarose (ChromoTek) was equilibrated in the resuspension buffer. Plant crude CCVs from CLC1-GFP and YFP were incubated with the GFP-Trap® Agarose for 2 hr at 4 °C. After, beads were washed three times with resuspension buffer. The RNA from purified CCVs was extracted using TRIzol Reagent (Invitrogen) following immunoisolation. Subsequently, the immunoisolated CCVs on GFP-Trap Agarose beads were further analyzed by a scanning electron microscope (SEM).

### Confocal microscopy
*Arabidopsis* seedlings were imaged on a Leica TCS SP5 confocal microscope (Leica Microsystems). Briefly, leaves were infiltrated with 10 μM FM4-64 dye for 30 mins before examination. Samples were imaged using a 40x water immersion dip-in lens mounted on a Leica TCS SP5 confocal microscope (Leica Microsystems). The excitation wavelength was 488 nm for GFP, 514 nm for YFP, and 559 nm for FM4-64. Emission was detected at 500–550 nm for GFP and YFP, 570–670 nm for FM4-64. The colocalization percentage was processed by means of the ImageJ (Fiji) software package. More specifically, the regions of interest (ROI) were selected to include vesicles accumulated around *B. cinerea* infection area. The total area of FM4-64 (red) and overlapped area (yellow) were measured and used for colocalization calculation. The average of three independent values was presented in the figures. The method for intracellular/PM signal intensity ratio quantification was described previously[55]. Non-saturated images were converted in ImageJ (Fiji) to 8-bit and regions of interest (ROIs) were selected based on the PM or cytosol localization. Intensity values per ROI of PM and intracellular signals were measured and used for calculation.

### Pathogen assay
*B. cinerea* was cultured on PDA plates for 10 days at room temperature before spores were collected and diluted into inoculation medium (0.1% sabouraud maltose broth buffer, Difco) for a final concentration of $10^5$ spores per mL. Four-week-old plants were then inoculated by applying a single 10 μl droplet of *B. cinerea* per leaf. Photos were taken 2 days after inoculation and analyzed by ImageJ software to measure the lesion size.

For pathogen assay using a mixture of WT or *Bc-dcl1/dcl2 B. cinerea* EVs with Δ*bcpls1* and *Bc-dcl1/dcl2*, $10^6$ *B. cinerea* spores were cultured for 48 hours at room temperature with shaking on the rocker in 100 ml of YEPD (0.3% yeast extract, 1% peptone, 2% glucose, pH 6.5) medium. *B. cinerea* EVs were isolated and purified using sucrose gradients as described above, then EVs from fraction four were resuspended in 100 μl PBS. The spores were collected and diluted into 2 x inoculation medium (0.2% sabouraud maltose broth buffer, Difco) for a final concentration of $2 \times 10^5$ spores per mL. Equal volumes of EVs and spores from mutant strains were mixed. Four-week-old plants were then inoculated by applying a single 10 μl droplet of EVs and spores mixture per leaf. Equal volumes of EVs and 2 x inoculation medium

were mixed to inoculate plants as the negative control. Photos were taken 2 days after inoculation and analyzed by ImageJ software to measure the lesion size.

### Real-time RT-PCR
Total RNA was extracted from plant tissues using TRIzol and the concentration was measured at 260 nm using a Nanodrop 2000 spectrophotometer (Nanodrop Technologies, Wilmington, DE). Equal amounts of total RNA were treated with DNase I enzyme (Roche, Basel, Switzerland) and converted into first-strand cDNA using SuperScript III (Invitrogen, Carlsbad, CA). The cDNA was then used to check target gene expression by real-time RT-PCR with gene-specific primers. Primer sequences are listed in Supplementary Data 1.

### Western blotting
Protein samples were resolved in a 12% SDS-polyacrylamide gel electrophoresis (PAGE) and then transferred to nitrocellulose membranes (GE Healthcare), blocked with 5% skimmed milk in TBST (Tris-buffered saline, 0.1% Tween 20) and probed with anti-GFP (Sigma-Aldrich), anti-mCherry (abcam) or anti-TET8. Then the signals were detected using the enhanced chemiluminescence method (GE Healthcare).

### AGO1 immunoprecipitation
*Arabidopsis* AGO1 IP was performed with 5 g fresh leaves collected 24 hr after spray inoculation with *B. cinerea*. Tissues were collected and ground in liquid nitrogen before resuspension in 5 ml of extraction buffer (20 mM Tris–HCl at pH 7.5, 150 mM NaCl, 5 mM MgCl₂, 1 mM DTT, 0.5% NP40, proteinase inhibitor cocktail; Sigma). The extract was then precleared by incubation with 50 ul of Protein A-agarose beads (Roche) at 4 °C for 30 min. The precleared extract was incubated with 10 ul of AGO1-specific antibody for 6 hours. 50 ul of Protein A-agarose beads (Roche) was then added to the extract before overnight incubation. Immunoprecipitates were washed three times (5 min each) in wash buffer (20 mM Tris–HCl at pH 7.5, 150 mM NaCl, 5 mM MgCl₂, 1 mM DTT, 0.5% Triton x-100, proteinase inhibitor cocktail; Sigma). RNA was extracted from the bead-bound AGO1 with TRIzol reagent (Invitrogen). Proteins were detected by boiling the bead-bound AGO1 in SDS-loading buffer before running in a SDS-PAGE gel.

### Scanning electron microscope (SEM)
The structure of CCVs isolated using GFP-Trap Agarose was examined using field emission scanning electron microscopy Nova NanoSEM 450 (FEI) at 15 kV. The samples were washed three times with PBS (pH 7.2) and fixed with 2.5% glutaraldehyde overnight at 4 °C. Samples were dehydrated with gradient alcohol solutions (10%, 30%, 50%, 70%, 80%, and 90%) for 10 min, 100% ethanol for 20 min, and 100% acetone for 30 min. Subsequently, samples were dried by $CO_2$ under critical point. Samples were sputter coated with gold and fixed on the stab using carbon tape before observation and imaging.

### Statistical analysis
The statistical analyses were performed using analysis of variance (ANOVA), Dunnett's multiple comparisons test, and unpaired two-tailed Student's t-test using GraphPad Prism (9.0.2). The statistical tests and n numbers, including sample sizes or biological replications, are described in the figure legends.

### Reporting summary
Further information on research design is available in the Nature Portfolio Reporting Summary linked to this article.

## Data availability
The data that support the findings of this study are available within the paper, Supplementary Information, and Source Data. Source data are provided with this paper.

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

## Acknowledgements

We thank Dr. Eugenia Russinova for providing CLC1-GFP and AP2A1-GFP seeds, Dr. Jiří Friml for providing the INTAM >> RFP-HUB1 and XVE >> AX2 lines, Shumei Wang for generating the *Arabidopsis* YFP-only transgenic line and Rachael Hamby and Lida Halilovic for editing the paper. This work was supported by National Institute of Health (R35GM136379), National Science Foundation (IOS 2020731), United State Department of Agriculture (2021-67013-34258), United States Department of Agriculture National Institute of Food and

Agriculture (2019-70016-29067) and the CIFAR 'Fungal Kingdom' fellowship to H. J., by National Natural Science Foundation of China (32272029), Hubei Provincial Natural Science Foundation of China (2022CFA079) to Q. C.

## Author contributions

H.J. conceived the idea and supervised the project. B.H. and H.J. designed the experiments. B.H. performed most of the experiments and analyzed data. H.W. developed key protocols and performed part of the data analysis. Q.C. validated mutant lines and performed pathogen assay. A. Chen performed the nanoparticle tracking and TEM analysis. A. Calvo generated constructs for *B. cinerea* transformation. G.L. generated the *flot1/2* double mutant lines. B.H., A.Chen, and H.J. wrote the manuscript.

## Competing interests

The authors declare no competing interests.
