## [Peer Review File · Nature Communications]

Fungal small RNAs ride in extracellular vesicles to enter plant cells through clathrin-mediated endocytosisREVIEWER COMMENTS

Reviewer #1 (Remarks to the Author):

The paper by He et al claim that the pathogen *Botrytis cinerea* secretes sRNAs via extracellular vesicles, and these sRNA then enter plant cells specifically via clathrin-mediated endocytosis (CME) to exert their effect upon the plant immune responses. Specifically, they show PLS is important in the EVs to secrete sRNAs and that these sRNA can be found in isolated plant CCVs.

While the work shows a solid logical progression, I feel this this work represents a refinement of the mechanism of material transfer rather than something truly novel. For example, they state that their pervious work shows that *Botrytis cinerea* can suppress plant immunity via sRNAs, and even reference papers which show that EVs are used in several models to secrete RNAs. Furthermore, CME is well known to be the dominant mode of extracellular material in plants, and there has been recent works showing that pathogens secrete materials which get taken up via CME (eg. Oliveira-Garcia et al, BiorXiv, 2022). There appear to be a number of critical experiments which lack repeats and quantification, things needed to show they are robust and reproducible findings. For example, Fig 1d, 2a, 4a-e, 5, 6 and supplemental 1c. And further, n numbers need to be clarified throughout. Eg, 1a – it states 4 replicates, but not actual numbers of measurements.

The use of Tyrphostin A23 as a specific CME inhibitor is a major problem. It does not work in the same way in plants as other model systems, so raises doubt it's a specific CME inhibitor (Dejonghe, et al, Nat comm, 2016). The authors must use alternative and up to date plant CME inhibitors, like ES9-19 – which appears to be the most specific plant CME blocker.

The authors use CME knockouts to show a reducing in sRNA uptake in plants. However, given that the mutants lines used have developmental defects and reduced FM uptake (which is also taken up by clathrin-independent pathways), the authors should consider using genetic inducible CME disruption lines (eg, CHC hub (Dhonukshe et al., 2007) auxilin-like (Ortiz-Morea et al., 2016)) to show a more specific effect.

If the authors want to claim that M β CD inhibits CIE, there should be some demonstration on its effect of endocytic flux. For example, FM uptake in the presence of M β CD – this would also give the readers an idea of how much CIE contributes to the overall uptake of extracellular materials. The same is true for the flot knockdown and rem mutant experiments. Figure 5 – In its current presentation of red and green, it is not accessible to colour blind readers. Further, the authors should determine a co-localisation value of the CCVs and the FM uptake vesicles. From the enlargement panel there does not appear to be a 100% overlap suggesting that there is also an enrichment of CIE at sites of infects, not just CME. It would be nice to see a similar experiment done with the CIE GFP-flot1 (which exists in the literature).

The purification of CCVs from infected plants is a very nice addition to the paper. However, I believe the authors should confirm that their isolated CCVs are intact (eg, negative staining of the CCV fraction).

b

Reviewer #2 (Remarks to the Author):

Fungal pathogens communicate with their plant host by the exchange of small RNAs. However, the precise mechanism of delivery is currently unknown. Here, the authors describe the interesting finding that small RNAs are transported within fungal EVs, which enter the plant cell through calthrin-meditated endocytosis. This was achieved by using the tetraspanin Pls1 as a molecular marker with this type of EVs. Candidate small RNAs were present in EVs and pls1 mutants were less virulent. Interestingly, virulence could be restored by the addition of wildtype EVs. Using an inhibitor for clathrin-mediated endocytosis virulence was affected suggesting that plant endocytosis is needed for successful infection.

Consistently, fungal small RNAs were loaded to a lower extent in plant AGO1 when plant mutants in clathrin-mediated endocytosis were tested. Also in these mutants the small RNA mediated reduction of target mRNAs was reduced. A cell biological analysis revealed that plant endocytosis is increased in cells adjacent to fungal hyphae. Finally, endocytotic vesicles were purified and fungal candidate small RNAs were present in these plant vesicles. This manuscript describes for the first time how fungal small RNAs enter the host plant. Therefore, it is of interest for a broad readership and merits publication in Nature Communication.

However, there are two major problems:

1. The cell biological analysis needs to be improved.

The main problem are the results presented in Figure 5. The authors use the lipophilic dye FM4-64 to stain vesicular structures in the vicinity of infected plant cells.

Previously, the authors published that these FM4-64 positive vesicles co-localise intensively with plant exosomes that are positive for the plant tetraspanin TET8-GFP.

See Figure 2 Cai et al 2018 Science.

In Figure 5 of the current publication the authors use FM4-64 staining to demonstrate intensive co-localisation of vesicles of potential fungal origin with plant endocytosis markers. However, FM4-64 should also stain plant exosomes (Cai et al 2018). Thus, currently it is unclear whether the FM4-64 stained vesicles are of plant or fungal origin. This needs to be addressed experimentally. The extent of co-localisation needs to be quantified and the previous data need to be revisited.

This also nicely illustrates, how difficult it is to draw conclusions from microscopic images. The same holds true for Figure 1. How do we know that the four signals in Figure 1 are EVs? Are such signals also seen in culture? How do their result compare to the previous study on Pls1-GFP from *Magnaporthe* (Clergeot et al. 2001) doi: 10.1073/pnas.111132998?

2. The EV complementation studies need to be improved:

It is unclear whether the complementary effects of EVs is solely based on the presence of Bc sRNAs. This could be addressed by applying Bc EVs from *dcl1/dcl2* double mutants with *delta bcpls1* strains. In comparison to wild type, EVs complementation should be significantly reduced.

Furthermore, it should be clarified, why *dcl1/dcl2* mutants are affected in virulence whereas Qin et al. 2022 doi: 10.1111/mpp.13269 reported no effects on virulence.

Reviewer #3 (Remarks to the Author):

The current manuscript by He and colleagues considers the mechanism by which miRNA are transferred between a pathogen and its plant host. The authors take a series of elegantly designed experiments to test several factors associated to the production and release of extracellular vesicles, the cargo of these EVs, as well as the mechanism by which they enter the plant cell. The authors have used several approaches that are complimentary and underscore the validity of the findings. Overall, an excellently executed project with some very convincing results in an area where we need to better understand the mechanism of transport of these key signalling nucleic acids.

I do have some minor critiques that would bear considering:

1. In the very beginning of the introduction, when discussing cross-kingdom sRNA, the authors have ignored the fact that this has been also shown in mutualistic systems. These studies should also be acknowledged.

2. In the second paragraph 'vasinfectum' should not be capitalized

3. Throughout the results the authors use terms like 'tried', 'attempted' etc. These should be

removed as the authors succeeded in each case and these words remove confidence from their results.

4. Throughout the results, the authors are very descriptive in their language and very rarely actually quote the values of the differences observed. I would appreciate for the authors to include values in the text supporting results for Fig 2, 3, 4.

5. The authors also do not typically discuss if their results are significant, nor do I see p-values given in the results text. This would be beneficial.

Reviewer #4 (Remarks to the Author):

It is my pleasure to review the manuscript NCOMMS-22-36782 by He et al. reporting on EV-mediated delivery of fungal small RNAs and their entry into plant cells mediated by clathrin-dependent endocytosis. The work performed in the manuscript is very extensive and very well written. This work is very exciting and potentially highly relevant. Therefore, it is important that the authors get it right and provide additional information to improve the quality of the manuscript:

1. The authors don't use MISEV guidelines although using NTA and TEM analysis. Please see <https://www.tandfonline.com/doi/full/10.1080/20013078.2018.1535750> For example, BcEV isolation/characterization was performed from conditioned media. Precautions such as the presence of particles in the media and percent dead cells should be considered and indicated. Please also provide experimental details following MISEV guidelines.

2. Related to point 1: The authors don't use sucrose gradients or SEC to improve the purity of the isolated BcEVs as well as EVs collected from plant apoplastic wash fluids, thereby reducing the possibility of e.g. co-pelleting protein aggregates.

Figure 1: Given the limited quality of the microscopic images, the authors cannot state that BcEVs were observed outside fungal cells. It is necessary to include e.g. plasmolysis experiments. Furthermore, co-localization of BcEVs with plant CCV markers as shown in Figure 5 as well as functional complementation of the FP-tagged proteins in deletion mutants are needed. Please show arrow bars for NTA measurements; and include a plant EV marker in the immunoblot analysis. A better quality immunoblot would be beneficial. What is the double band?

Figure 2: Can the authors show (by NTA, TEM) that EVs are intact after MNase treatment and ruptured with Triton-x-100? Regarding the Bc deletion strains, please show at least two independent mutants for each gene and confirmation of genetic knock-out by RT-PCR, for example. For phenotyping, please include growth curves and growth on minimal medium as well as quantification of BcEV production across the strains. In figure 2c, were similar amounts of BcEVs collected across the strains and used for RT-PCR? Please also include siR17. Figure 2d, if the authors expose plants to BcEVs as performed in the trans-complementation experiments, it is necessary to understand whether BcEVs would induce any responses in plants.

Figure 3: It is surprising that TyrA23 does not affect Bc growth and sporulation. Can the authors explain and show that the treatment does inhibit endocytosis e.g. by FM4-64 tracing? Is it possible that the lesions in figure 3b pictures appear not fully matching the quantitative analysis of chc2? And given the much smaller leaves, how informative are results obtained from the ap2 σ mutant?

Figure 4: Are all mutants expressing similar levels of AGO1? Is it correct that all experiments were done from BcEV-treated plants but figure 4f shows results from Bc-infected plants?

Figure 5: Related to figure 1, please show co-localization of BcPSL1-YFP with plant CCV markers.

Figure 6: It is known that some Arabidopsis proteins have per se an affinity for GFP, therefore, is IgG a suitable control? Please show TEM of CCV-pull-down to confirm the presence of CCVs.

Discussion: How do the authors envisage the uptake process of BcEVs into plants and then the binding of Bc small RNAs to plant AGO1? Is BcPSL1 present at CCVs? Is AGO1 present at CCVs? Or are vesicles ruptured to release the Bc small RNAs?

Methods: I do not understand the sentences "RNA was extracted from the bead-bound EVs with TRIZOL reagent (Invitrogen). Proteins were detected by boiling the bead-bound EVs in SDS-loading buffer before running in a SDS-PAGE gel." In the context of the AGO1 IPs.

REVIEWER COMMENTS

Reviewer #1 (Remarks to the Author):

The paper by He et al claim that the pathogen *Botrytis cinerea* secretes sRNAs via extracellular vesicles, and these sRNA then enter plant cells specifically via clathrin-mediated endocytosis (CME) to exert their effect upon the plant immune responses. Specifically, they show PLS is important in the EVs to secrete sRNAs and that these sRNA can be found in isolated plant CCVs.

While the work shows a solid logical progression, I feel this this work represents a refinement of the mechanism of material transfer rather than something truly novel. For example, they state that their pervious work shows that *Botrytis cinerea* can suppress plant immunity via sRNAs, and even reference papers which show that EVs are used in several models to secrete RNAs. Furthermore, CME is well known to be the dominant mode of extracellular material in plants, and there has been recent works showing that pathogens secrete materials which get taken up via CME (eg. Oliveira-Garcia et al, BiorXiv, 2022).

Response: We appreciate the reviewer's feedback. We've previously shown that *B. cinerea* can send small RNA (sRNA) to plant cells and silence host genes to promote infection¹. But how fungal sRNAs travel through the extracellular environment and survive numerous ribonucleases in the extracellular space remains unknown. Furthermore, it is not clear how fungal sRNAs enter the plant cells to silence plant genes. This manuscript has addressed these two questions and discovered that *B. cinerea* uses EVs to protect and transport sRNAs, which enter plant cells mainly through the clathrin-mediated endocytosis (CME). Although EVs were reported in various fungi, but no reports have yet demonstrated the involvement of fungal EVs in cross-kingdom RNA trafficking. Our study has provided the first example of fungal EV-mediated cross-kingdom sRNA trafficking.

Similarly, as the reviewer pointed out (Oliveira-Garcia et al. BioRxiv, 2022), two recent studies in The Plant Cell report that pathogen protein effectors are taken up by plants via CME^{2,3}. But these studies neither mentioned EVs nor small RNAs in CME-mediated uptake. These protein effectors are most likely not associated with EVs because they have signal peptide for conventional secretion. Therefore, our work is likely the first to show that fungal EVs carry small RNAs, which are taken up by plant CME pathway.

There appear to be a number of critical experiments which lack repeats and quantification, things needed to show they are robust and reproducible findings. For example, Fig 1d, 2a, 4a-e, 5, 6 and supplemental 1c. And further, n numbers need to be clarified throughout. Eg, 1a – it states 4 replicates, but not actual numbers of measurements.

Response: We have included two additional replicates for all blots and gel images in the supplementary Fig.13. Additionally, we have also included replicates of confocal images in Figure 5 and Figure 6 in the supplementary Fig. 13. The n numbers are now clarified in all of the figure legends.

The use of Tyrphostin A23 as a specific CME inhibitor is a major problem. It does not work in the same way in plants as other model systems, so raises doubt it's a specific CME inhibitor (Dejonghe, et al, Nat comm, 2016). The authors must use alternative and up to date plant CME inhibitors, like ES9-19 – which appears to be the most specific plant CME blocker.

Response: Thanks a lot for pointing this out, we highly appreciate that. We tried the specific CME inhibitor ES9-17⁴ and observed that ES9-17 strongly suppressed CME. However, it also inhibits the growth of *B. cinerea* (Supplementary Fig. 8b). Therefore, we, we decided to focus on genetic mutants instead of chemical inhibitor-treated plants when investigating the plant-fungi interaction, and included new genetic data using inducible CME disruption Arabidopsis lines INTAM>>RFP-HUB1 and XVE>>AX2. The results are presented in Figure 3b, c.

The authors use CME knockouts to show a reducing in sRNA uptake in plants. However, given that the mutants lines used have developmental defects and reduced FM uptake (which is also taken up by clathrin-independent pathways), the authors should consider using genetic inducible CME disruption lines (eg, CHC hub (Dhonukshe et al., 2007) auxilin-like (Ortiz-Morea et al., 2016)) to show a more specific effect.

Response: We thank the reviewer for this excellent suggestion which can further prove the function of CME in sRNA uptake in plants. Prof. Jiří Friml generously provided us with the inducible CME disruption Arabidopsis lines INTAM>>RFP-HUB1 and XVE>>AX2, which can block plant CME after induction. Using these lines, we observed similar results to the *chc2* and *ap2* mutants where once CME was blocked, the plants exhibited greater resistance to *B. cinerea* infection (Fig. 3b,c). Furthermore, the induced lines also reduced the amount of *B. cinerea* sRNAs loaded into plant AGO1 (Fig. 4b,c). In addition, the silencing of target genes of Bc-sRNA effectors were reduced in the induced CME disruption lines (Supplementary Fig. 11). These results further support that CME is important for Bc-sRNA internalization into plant cells.

If the authors want to claim that M β CD inhibits CIE, there should be some demonstration on its effect of endocytic flux. For example, FM uptake in the presence of M β CD – this would also give the readers an idea of how much CIE contributes to the overall uptake of extracellular materials. The same is true for the flot knockdown and rem mutant experiments.

Response: Considering the broad impact of chemical inhibitor treatments on both plants and fungi, the results obtained from such experiments may not accurately reflect the real plant-fungi interaction conditions. So in the revised manuscript, we mainly focus on the mutant plants of both pathways when studying the

plant-fungi interaction. However, to gain a general understanding of the roles played by CME and CIE pathways in the uptake of extracellular materials by plants, we measured the FM4-64 internalization following the treatment of *Arabidopsis* by CME inhibitor ES9-17 or CIE inhibitor M β CD (Supplementary Fig 8a, c). The results showed that the FM4-64 uptake decreased by about 21% after M β CD treatment. While the ES9-17 treatment can suppress the FM4-64 uptake by 71% compared with the control treatment. This result indicates that CME plays a more important role in taking up extracellular materials. Because there is a close homolog of *Flot1* in *Arabidopsis* named *Flot2*, we generated the knockout mutant of *Flot2* in the *flot1* mutant background using the Crispr Cas9 system. We found that both *flot* and *rem* mutants only showed a slight decrease in FM4-64 internalization (Supplementary Fig. 9d).

Figure 5 – In its current presentation of red and green, it is not accessible to colour blind readers. Further, the authors should determine a co-localisation value of the CCVs and the FM uptake vesicles. From the enlargement panel there does not appear to be a 100% overlap suggesting that there is also an enrichment of CIE at sites of infects, not just CME. It would be nice to see a similar experiment done with the CIE GFP-*flot1* (which exists in the literature).

Response: We thank the reviewer for the thoughtful suggestion. As recommended, we have made changes to all our confocal images and used color-blind-friendly colors. Additionally, we have conducted further analysis to measure the colocalization between CCVs and FM 4-64 stained vesicles. The degree of colocalization between FM4-64 and the CCV signals from AP2A1-GFP, CLC1-GFP, and CHC2-YFP was determined to be 46%, 52%, and 60%, respectively (Fig. 5). However, as FM4-64 can stain all membrane structures, including the plasma membrane, extracellular vesicles, and endocytic vesicles, it remains difficult to tell the exact percentage of CIE and CME in the infection area based on the colocalization percentage between FM4-64 and CME makers, because we can't distinguish the FM4-64-stained extracellular vesicles signals (from both plants and fungi) from the endocytic vesicle signals. So we generated an *Arabidopsis* YFP-FLOT1 transgenic line and inoculated it with wild-type *B. cinerea*. Our results showed that during *B. cinerea* infection, only a small fraction (10.5%) of FLOT1 labeled vesicles co-localized with FM 4-64 stained vesicles (Fig. 5d). These findings suggest that the CME pathway plays a more important role than CIE in sRNA endocytosis during *B. cinerea* infection.

The purification of CCVs from infected plants is a very nice addition to the paper. However, I believe the authors should confirm that their isolated CCVs are intact (eg, negative staining of the CCV fraction).

Response: We appreciate the reviewer's remarks about the inclusion of the purification of CCVs from infected plants. We agree with the reviewer that it is essential to ensure the integrity of the isolated CCVs. We now included the scanning electron microscopy (SEM) examination of the immunopurified CCVs and have

confirmed that most of the isolated CCVs are intact. As shown in Fig. 6b (copied below), many intact CCVs with typical cage structures can be seen after immune-purification from plant cells.

Fig. 6b from the revised manuscript.

b, Intact CCVs can be observed on the surface of GFP-Trap Agarose beads by SEM after CCV immune-purification from CLC1-GFP plants. Scale bar, 100 nm.

Reviewer #2 (Remarks to the Author):

Fungal pathogens communicate with their plant host by the exchange of small RNAs. However, the precise mechanism of delivery is currently unknown. Here, the authors describe the interesting finding that small RNAs are transported within fungal EVs, which enter the plant cell through clathrin-mediated endocytosis. This was achieved by using the tetraspanin Pls1 as a molecular marker with this type of EVs. Candidate small RNAs were present in EVs and pls1 mutants were less virulent. Interestingly, virulence could be restored by the addition of wildtype EVs. Using an inhibitor for clathrin-mediated endocytosis virulence was affected suggesting that plant endocytosis is needed for successful infection. Consistently, fungal small RNAs were loaded to a lower extent in plant AGO1 when plant mutants in clathrin-mediated endocytosis were tested. Also in these mutants the small RNA mediated reduction of target mRNAs was reduced. A cell biological analysis revealed that plant endocytosis is increased in cells adjacent to fungal hyphae. Finally, endocytotic vesicles were purified and fungal candidate small RNAs were present in these plant vesicles.

This manuscript describes for the first time how fungal small RNAs enter the host plant. Therefore, it is of interest for a broad readership and merits publication in Nature Communication.

However, there are two major problems:

1. The cell biological analysis needs to be improved.

The main problem are the results presented in Figure 5. The authors use the lipophilic dye FM4-64 to stain vesicular structures in the vicinity of infected plant cells. Previously, the authors published that these FM4-64 positive vesicles co-localise intensively with plant exosomes that are positive for the plant tetraspanin TET8-GFP. See Figure 2 Cai et al 2018 Science.

In Figure 5 of the current publication the authors use FM4-64 staining to demonstrate intensive co-localisation of vesicles of potential fungal origin with plant endocytosis markers. However, FM4-64 should also stain plant

exosomes (Cai et al 2018). Thus, currently it is unclear whether the FM4-64 stained vesicles are of plant or fungal origin. This needs to be addressed experimentally. The extent of co-localisation needs to be quantified and the previous data need to be revisited.

Response: We thank the reviewer for the comments and suggestions. We fully agree with the reviewer's comments that FM4-64 has the capability to stain all membrane structures, including the plasma membrane, EVs, and endocytic vesicles. Fig.5 mainly demonstrates that the plant CME pathway can be activated during *B. cinerea* infection as the clathrin coat component proteins CLC1, CHC2, and adaptor protein AP2A1 are all found to be accumulated around the infection area. To confirm the function of CME in *B. cinerea* EVs uptake, we performed a new experiment to infect CLC1-GFP plants with the *B. cinerea* BcPLS1-mCherry strain. We found that plant CLC1 protein colocalizes with *B. cinerea* EV marker protein BcPLS1 well (Fig. 6c) and BcPLS1 can be detected in the isolated plant CCVs (Fig. 6d). These results provide direct evidence that *B. cinerea* EVs can be internalized through plant CME pathway.

This also nicely illustrates, how difficult it is to draw conclusions from microscopic images. The same holds true for Figure 1. How do we know that the four signals in Figure 1 are EVs ? Are such signals also seen in culture ? How do their result compare to the previous study on Pls1-GFP from Magnaporthe (Clergeot et al. 2001) doi: 10.1073/pnas.111132998?

Response: We have revised Figure 1 and included additional evidence to further characterize the *B. cinerea* EVs.

First, after collecting *B. cinerea* EVs from the culture using 100, 000 x g ultracentrifugation, we further fractionated the EVs using sucrose gradients. We found that most of the particles are concentrated in fractions 4 and 5 with a density of 1.15 to 1.19 gml⁻¹, which is consistent with the density of exosomes in animal and plant systems. The peak diameters of fractions 4 and 5 are 93nm and 94nm, respectively (Fig. 1a). In addition, TEM results showed that typical cup-shaped EV structures could be observed in both fractions 4 and 5 (Fig. 1b). Western blot and confocal microscopy results also support that the *B. cinerea* EV marker protein BcPLS1 is mainly present in fraction 4 (Fig. 1c). The BcPLS1-YFP fluorescent signal can be observed from EVs isolated from BcPLS1-YFP liquid culture (Fig. 1d). The results on EVs from cultured *B. cinerea* are copied below.

Fig.1a to d from the revised manuscript. *B. cinerea* secretes EVs in culture.

a, The size distribution of the isolated EVs in fractions 4 and 5 was measured using nanoparticle tracking analysis. The data of 4 measurements in one replicate was presented here. Similar data were obtained in three biological replicates. The red area represents the error bars. b, Transmission electron microscopy negative stain images of fractions 4 and 5. Scale bar, 100nm. c, Western blot analysis of BcPLS1 and BcTSP3 in sucrose gradient fractions. d, EVs isolated from BcPLS1-YFP and BcTSP3-YFP liquid culture were examined by confocal microscopy. Scale bar, 10 μ m.

Second, to further prove that *B. cinerea* can secrete EVs during infection, we infected *Arabidopsis* plants with the *B. cinerea* BcPLS1-YFP strain and collected EVs from the infected plants. We found that both plant exosomes labeled by TET8 and fungal BcPLS1-YFP-positive EVs were concentrated in the same fractions after sucrose separation (Fig. 1f). We further infected *Arabidopsis* TET8-CFP transgenic plants with *B. cinerea* BcPLS1-YFP strain and collected EVs from infected plants. Two separate groups of EVs with different fluorescent signals can be observed under confocal microscopy (Fig. 1g). These results can support that *B. cinerea* can secrete EVs during infection.

Fig. 1e to g from the revised manuscript. *B. cinerea* can secrete BcPLS1-positive EVs during infection. e, BcPLS1-YFP labeled EVs were observed outside the fungal cells at the site of infection on Arabidopsis leaves. Samples were pretreated with 0.75M sorbitol for 30min to induce plasmolysis before imaging. Scale bars 10 μ m. f, BcPLS1-YFP and BcTSP3-YFP were detected in sucrose gradient fractionated EVs that were isolated from BcPLS1-YFP or BcTSP3-YFP infected wild-type Arabidopsis using western blot. Arabidopsis TET8 was used as a plant exosome control. g, EVs isolated from BcPLS1-YFP or BcTSP3-YFP infected Arabidopsis TET8-CFP plants were examined using confocal microscopy. Scale bars 10 μ m.

The previous study about PLS1 in *Magnaporthe grisea* showed that the *pls1* mutant could not penetrate host leaf surfaces and that PLS1-GFP is localized to appressorial membranes during infection, which is consistent with our results. We also observed localization of PLS1-YFP to the appressorial membrane in *B. cinerea* (Fig. 1e). We further found that the PLS1 protein is also localized on EVs. The amount of EVs isolated from *B. cinerea pls1* mutant was reduced significantly compared to the wild-type strain (Supplementary Fig. 6b), which affects the secreted Bc-sRNAs. Moreover, the decreased secretion of *B. cinerea* EVs observed in the *pls1* mutant may also impair the delivery of fungal penetration-related enzymes, which could explain the mutant's deficient penetration phenotype of *Magnaporthe*.

2. The EV complementation studies need to be improved:

It is unclear whether the complementary effects of EVs is solely based on the presence of Bc sRNAs. This could be

addressed by applying Bc EVs from *dcl1/dcl2* double mutants with delta *bcpls1* strains. In comparison to wild type, EVs complementation should be significantly reduced.

Response: Thanks for your suggestion. To confirm the function of Bc sRNAs in fungal EVs, we have performed the suggested complementation experiment using *Bc-dcl1/dcl2* mutant EVs. As shown in supplementary Fig.7, the *Bc-dcl1/dcl2* mutant EVs showed reduced complementation of the *Δbcpls1* mutant virulence compared with wild-type EVs. However, complementation with the *Bc-dcl1/dcl2* EVs still slightly recovered the virulence of *Δbcpls1* when compared with the *Δbcpls1* mutant itself. This is likely due to the presence of other cargoes in fungal EVs, such as virulence-related proteins and/or metabolites.

Furthermore, it should be clarified, why *dcl1/dcl2* mutants are affected in virulence whereas Qin et al. 2022 doi: 10.1111/mpp.13269 reported no effects on virulence.

Response: We have submitted a manuscript in response to the Qin et al. study to Molecular Plant Pathology, which is currently under review. We also deposited it to BioRxiv, which is available at <https://www.biorxiv.org/content/10.1101/2022.12.30.522274v3>.

In brief, we believe one major reason why Qin et al. didn't observe the same effect of their *dcl1dcl2* mutant on virulence as we did is that the *dcl1dcl2* mutant they generated was in the *ku70* mutant background instead of wild type background by CRISPR CAS approach. This *B. cinerea ku70* mutant strain is compromised in the non-homologous end-joining DNA repair (NHEJ) system. It's well known that the CAS9 protein can generate gRNA-guided cut and unpredicted off-target cleavages^{5,6}, which largely rely on NHEJ DNA repair system. Using CRISPR-CAS in the NHEJ mutant background likely damage the genome integrity and also cause unpredicted DNA damage. Furthermore, double deletion of NHEJ and DCLs genes should be characterized with extra caution because RNAi mutants including DCLs, showed increased sensitivity to DNA damage in fungi, such as in *Neurospora crassa*⁷, a phenomenon well-studied in the mammalian field as well⁸⁻¹⁰. In addition, a *ku70* mutant may alter fungal pathogenicity. For example, the *ku70* mutant of the fungus *Penicillium digitatum* exhibited altered growth and conidia production¹¹. Therefore, NHEJ plus DCLs deletion likely causes unpredictable complications regarding DNA repair, genome integrity and background DNA damage and mutations. Indeed, principal component analysis (PCA) showed that the sRNA profiles of the *ku70* mutant have big differences from that in wild type *B. cinerea* B05.10 strain (Figure 1a).

Furthermore, for the *dcl1/dcl2* infection assay, Qin et al. used a relatively high spore concentration of 1000 conidia/μl suspended in potato dextrose broth (PDB), a rich culture medium that allows rapid growth of *B. cinerea in vitro*, to inoculate the detached leaves. It is worth noting that tomato is a much more favored host for *B. cinerea* than Arabidopsis. We often use 10-100 fold less inoculon for tomato than for Arabidopsis (Weiberg et al, 2013). We have repeated the same infection assay using the Qin et al.'s *bcdcl1/bcdcl2/ku70*

strain with a lower amount of spores (10 μ l of 10 conidia/ μ l) in 1% malt extract medium. We observed reduced virulence in the *bcdcl1/bcdcl2/ku70* mutant strains at 60 hours post-inoculation (Figure 1b). This lower spore concentration better represents the natural infection pressure.

Figure 1. Infection assay of *Abcdcl1/Abcdcl2/ku70* on detached tomato leaves and PCA analysis of *ku70*, *dcl1dcl2*, and wild-type B05.10 strains.

(a) The small RNA sequencing libraries of the *ku70* mutant, *dcl1dcl2* mutant strain, and wild-type B05.10 strain were subjected to PCA analysis. (b) Inoculation was performed with 100 conidiospores per inoculation, and lesion areas were analyzed at 60 hours post-inoculation (hpi). At least 30 lesions were measured per strain and condition, and inoculation assays were repeated three times with similar results. Letters indicate significant differences when using one-way ANOVA and Tukey test with $p < 0.05$.

We would like to emphasize that the Botrytis *dcl1*, *dcl2* single mutants and *dcl1dcl2* double mutant strains we generated previously are all in the wild type B05.10 Botrytis strain background. We found that the biogenesis of fungal small RNA effectors largely relies on both Botrytis DCL1 and DCL2, the *dcl1dcl2* double mutant displayed reduced fungal growth and was less virulent on Arabidopsis, tomato, grapes lettuce and rose (See Figure 2a,b) from previous publications^{1,12}. Single mutants of *dcl1* or *dcl2* also displayed growth defect but did not have much virulence defect because they have redundancy in the biogenesis of small RNA effectors (Figure 2b).

Figure 2. *B. cinerea* dcl1dcl2 double mutant, but not the dcl1 or dcl2 single mutants, displays reduced virulence on fruits, vegetables, and flower petals. a, *B. cinerea* dcl1dcl2 double mutant shows strongly compromised virulence on fruits (tomato, strawberry and grape), vegetables (lettuce and onion) and flower petals (rose), and *B. cinerea* dcl1 and dcl2 single mutants showed similar virulence as the WT strain. b, *B. cinerea* dcl1dcl2 double mutant showed stronger growth retardation than dcl1, dcl2 single mutants. Bc-sRNA can't be detected in the dcl1dcl2 double mutant but can be detected in the dcl1 and dcl2 single mutants.

The important role of Botrytis DCL1/DCL2 in infection was also observed by several other labs when they knocked down (not even knockout) DCL1/DCL2 genes by silencing. Below are two such examples:

1. Minicell-based fungal RNAi delivery for sustainable crop protection: <https://ami-journals.onlinelibrary.wiley.com/doi/full/10.1111/1751-7915.13699>

Md Tabibul Islam, Zachery Davis, Lisa Chen, Jacob Englaender, Sepehr Zomorodi, Joseph Frank, Kira Bartlett, Elisabeth Somers, Sergio M. Carballo, Mark Kester, Ameer Shakeel, Payam Pourtaheri, Sherif M. Sherif.

In this study, the authors demonstrated that minicell-encapsulated dsRNA (ME-dsRNA) of DCL genes knocked down (not even knockout) Botrytis DCL1 and DCL2, led to significant fungal growth inhibition and halted disease progression in strawberries for 12 days (Fig.4 that copied below).

2. Double-stranded RNA targeting fungal ergosterol biosynthesis pathway controls *Botrytis*

cinerea and postharvest grey mould. <https://onlinelibrary.wiley.com/doi/full/10.1111/pbi.13708>

This study reported reduction in disease on onion, rose and strawberry after application of dsRNA targeting *Botrytis* DCL genes (*dcl1* and *dcl2*), which knockdown both DCLs. This was used as a positive control.

Mutation of *DCL* genes in fungi can significantly attenuate virulence of other fungal pathogens. For example, knocking out of *DCL* genes from *Colletotrichum gloeosporioides*¹³, *Fusarium graminearum*¹⁷, and *Valsa mali*¹⁵ can significantly reduce the fungal pathogenicity on plants. Except for knocking out, knocking down the expression of *DCL* using RNAi can affect the virulence of *Penicillium italicum* on citrus fruit¹⁴.

Furthermore, spray-induced gene silencing (SIGS) has emerged as a promising approach to combat fungal infections in plants. Using SIGS to target fungal DCL genes has been shown to effectively protect a wide variety of vegetables, fruits, and crops from fungal infections. Externally applied Bc-DCL1/2 dsRNAs can inhibit *B. cinerea* infection in fruits such as tomatoes, strawberries, and grapes, as well as in vegetables like lettuce and onion, rose petals, and chickpeas^{12,18-20}. Moreover, spraying of dsRNA that target *F. graminearum* DCL genes can significantly improve the Barley resistance to *F. graminearum* infection¹⁶. Similarly, dsRNAs that target *Plasmopara viticola* DCL genes have been demonstrated to control the downy mildew disease on grapevines²¹. Targeting fungal DCL genes using SIGS has been developed into an effective means for controlling fungal diseases, underscoring the important role of DCL genes in fungal virulence.

The phenomenon of cross-kingdom RNAi has been widely observed in many different kinds of interacting organisms, including plant fungal pathogens (such as *Fusarium oxysporum* and *Verticillium dahliae*), oomycete pathogens (such as *Hyaloperonospora arabidopsidis*), the parasitic plant *Cuscuta campestris*, and even the symbiotic ectomycorrhizal fungus *Pisolithus microcarpus*, the bacterial symbiont *Rhizobium* (*Bradyrhizobium japonicum*)^{12,22-27}. Cross-kingdom RNAi has also been observed between animals and interacting parasites and pathogens. Furthermore, the mode of action of transferred microbial sRNAs in the hosts seems to be conserved. The sRNAs from *F. oxysporum*²⁶, *V. dahliae*¹², *H. arabidopsidis*²² and the *Rhizobium*²⁴ all utilize host AGO proteins for silencing host target genes. This mechanism is even conserved in the interaction between mosquitoes and the fungal pathogen, *Beauveria bassiana*. *B. bassiana* transfers a microRNA-like RNA (miRNA) to the host cells and loads the fungal miRNA into mosquito AGO1 to silence the host immunity gene *Toll receptor ligand Spätzle 4*²⁸. We have recently demonstrated that *Botrytis* small RNAs could be pulled down by tomato AGO1 antibody indicating that cross-kingdom RNAi is also present in tomato *Botrytis* interaction.

Reviewer #3 (Remarks to the Author):

The current manuscript by He and colleagues considers the mechanism by which miRNA are transferred between a pathogen and its plant host. The authors take a series of elegantly designed experiments to test several factors associated to the production and release of extracellular vesicles, the cargo of these EVs, as well as the mechanism by which they enter the plant cell. The authors have used several approaches that are complimentary and underscore the validity of the findings. Overall, an excellently executed project with some very convincing results in an area where we need to better understand the mechanism of transport of these key signalling nucleic acids.

I do have some minor critiques that would bear considering:

1. In the very beginning of the introduction, when discussing cross-kingdom sRNA, the authors have ignored the fact that this has been also shown in mutualistic systems. These studies should also be acknowledged.

Response: We thank the reviewer for bringing to our attention that we inadvertently omitted information about cross-kingdom RNAi in mutualistic systems from the manuscript. We apologize for this mistake and have revised the manuscript to include the relevant information.

2. In the second paragraph 'vasinfectum' should not be capitalized

Response: Thanks for pointing it out. It has been corrected in the revised manuscript.

3. Throughout the results the authors use terms like 'tried', 'attempted' etc. These should be removed as the authors succeeded in each case and these words remove confidence from their results.

Response: Many thanks for pointing this out. We have removed such language and instead used more definitive language to accurately convey our findings.

4. Throughout the results, the authors are very descriptive in their language and very rarely actually quote the values of the differences observed. I would appreciate for the authors to include values in the text supporting results for Fig 2, 3, 4.

Response: We appreciate the reviewer's suggestion to include values in the text. We have revised the manuscripts and added the values to the text based on your suggestion.

5. The authors also do not typically discuss if their results are significant, nor do I see p-values given in the results text. This would be beneficial.

Response: We thank the reviewer's suggestion. In the revised manuscript, p-values have been added to the figures and the significant differences have been discussed in the manuscript.

Reviewer #4 (Remarks to the Author):

It is my pleasure to review the manuscript NCOMMS-22-36782 by He et al. reporting on EV-mediated delivery of fungal small RNAs and their entry into plant cells mediated by clathrin-dependent endocytosis. The work performed in the manuscript is very extensive and very well written. This work is very exciting and potentially highly relevant. Therefore, it is important that the authors get it right and provide additional information to improve the quality of the manuscript:

1. The authors don't use MISEV guidelines although using NTA and TEM analysis. Please see <https://www.tandfonline.com/doi/full/10.1080/20013078.2018.1535750>

For example, BcEV isolation/characterization was performed from conditioned media. Precautions such as the

presence of particles in the media and percent dead cells should be considered and indicated. Please also provide experimental details following MISEV guidelines.

Response: Thank you for your suggestion. We have taken it into consideration and made changes to our revised manuscript accordingly. We now follow the MISEV guidelines to characterize EVs isolated from *B. cinerea* culture and Botrytis-infected plant apoplastic fluids in the revised version. Specifically, to isolate EVs from the *B. cinerea* culture, we first pre-clarified the medium by ultracentrifugation at 100,000xg and then filtration through a 0.2 μm filter to remove any possible particles. The pre-clarified medium was used to culture *B. cinerea* cells. We then cultured the fungus and centrifuged the cultured fungal cells at a low speed to pellet the cells and removed any possible remaining cells in the culture by filtering the supernatant again through a 70 μm cell strainer. After the initial steps to remove cells and other large particles, the supernatant was subjected to further centrifugation at 10,000xg to remove any remaining small particles and broken cells. To ensure the purity of the supernatant, we filtered it again through a 0.45 μm filter. The filtered supernatant was then ultracentrifuged at 100,000xg to pellet the crude EVs. The crude EVs were subsequently purified by centrifugation through 10%-90% sucrose gradients. Our results showed that *B. cinerea* EVs were mainly enriched at the density of 1.11–1.19 g ml^{-1} , which is similar to animal and plant exosomes. Detailed information was added to the Methods section.

2. Related to point 1: The authors don't use sucrose gradients or SEC to improve the purity of the isolated BcEVs as well as EVs collected from plant apoplastic wash fluids, thereby reducing the possibility of e.g. co-pelleting protein aggregates.

Response: We used sucrose gradients to further purify EVs isolated from both *B. cinerea* liquid culture and Botrytis-infected plants. After sucrose gradient separation, we found that fungal EVs are mainly observed in fractions 4 and 5 (Fig. 1b). In addition, the Western blot indicated that the *B. cinerea* EV marker protein BcPLS1 was also concentrated mainly in fraction 4 & 5 (Fig. 1c). We also detected the sRNAs in *B. cinerea* EVs after sucrose gradient separation, and we found that the sRNAs secreted by *B. cinerea* were also concentrated in the fraction 4 & 5 (Fig. 2a). These results further indicate that *B. cinerea* uses EVs to deliver sRNAs.

Figure 1: Given the limited quality of the microscopic images, the authors cannot state that BcEVs were observed outside fungal cells. It is necessary to include e.g. plasmolysis experiments. Furthermore, co-localization of BcEVs with plant CCV markers as shown in Figure 5 as well as functional complementation of the FP-tagged proteins in deletion mutants are needed. Please show arrow bars for NTA measurements; and include a plant EV marker in the immunoblot analysis. A better quality immunoblot would be beneficial. What is the double band?

Response: We appreciate the reviewer’s valuable comments. We performed plasmolysis experiments after the BcPLS1-YFP Botrytis strain inoculation, and indeed we can observe much better EV signals around the infection area (Fig. 1e in the manuscript and copied above in response to reviewer 2).

To further confirm that *B. cinerea* can secrete EVs during infection, we inoculated the Arabidopsis TET8-CFP line with *B. cinerea* BcPLS1-YFP strain and isolated EVs from infected plant apoplastic fluids. Two different groups of EVs with distinct signals were observed. One group was labeled with the TET8-CFP signal, which indicates the plant exosomes, while another group of EVs was labeled with the BcPLS1-YFP signal, which indicates *B. cinerea* EVs during infection (Fig. 1g in the manuscript, copied above in response to Reviewer2). Sucrose gradient fractionation was used to further purify EVs isolated from BcPLS1-YFP strain-infected wild-type plants. We found that both BcPLS1 and TET8 were concentrated in the same fractions, which further proves *B. cinerea* can secrete EVs when infecting plants (Fig. 1f in the manuscript copied above in response to reviewer 2).

The confocal results of colocalization between BcEVs with plant CCVs have been added to the revised Fig. 6c. We further purified CCVs using immunoprecipitation from BcPLS1-mCherry infected CLC1-GFP plants, the BcPLS1 protein can be detected in the purified plant CCVs fractions (Fig. 6c, d, copied below).

Fig. 6c and d from the revised manuscript.

c, BcPLS1-mCherry colocalize with Arabidopsis CLC1-GFP during infection. The BcPLS1-mCherry strain was used to infect Arabidopsis CLC1-GFP transgenic plant. Confocal pictures were taken after ten hours of infection. Scale bar, 10 μ m. **d**, BcPLS1-mCherry signals can be detected in GFP-Trap Agarose beads-isolated CCVs from BcPLS1-mCherry infected CLC1-GFP plants using western blot. YFP plant was used as a negative control.

We generated complementary strains by expressing BcPLS1-CFP and BcTSP3-CFP in the *Abcpls1* and *Abctsp3* mutant strains, respectively. Pathogenicity assays demonstrated that both complementary strains restored the pathogenicity (Fig 2c and Supplementary Fig. 4). The red shading in the NTA results indicates standard error. Finally, a higher quality immunoblot has been included in the revised version. Please see

Supplementary Fig. 2c. Sometimes, the double band of BcPLS1-YFP can be observed, which may represent the modification of BcPLS1. Actually, many tetraspanin proteins have post-translational modifications, including N-linked glycosylation on the EC2 loop and palmitoylation at a CXXC motif in their transmembrane region in the mammalian system³⁰. The exact modification of BcPLS1 will require further study though.

Figure 2: Can the authors show (by NTA, TEM) that EVs are intact after MNase treatment and ruptured with Triton-x-100? Regarding the Bc deletion strains, please show at least two independent mutants for each gene and confirmation of genetic knock-out by RT-PCR, for example. For phenotyping, please include growth curves and growth on minimal medium as well as quantification of BcEV production across the strains. In figure 2c, were similar amounts of BcEVs collected across the strains and used for RT-PCR? Please also include siR17. Figure 2d, if the authors expose plants to BcEVs as performed in the trans-complementation experiments, it is necessary to understand whether BcEVs would induce any responses in plants.

Response: We thank the reviewer's comments. The results obtained from TEM analysis demonstrated that even after MNase treatment, the fungal EVs remained intact. Upon the addition of Triton X-100 to the reaction, all vesicles were ruptured (Supplementary Fig. 3).

Supplementary Fig. 3a

Transmission electron microscopy analysis demonstrated that *B. cinerea* extracellular vesicles (EVs) remained intact following MNase treatment. However, the addition of Triton X-100 to the reaction led to the rupture of *B. cinerea* EVs. Scale bar, 100nm.

Pathogenicity assay and RT-PCR results of two independent deletion strains and complementary strains have been included in Fig. 2c and Supplementary Fig. 4. The growth curves of $\Delta bcpls1$, $\Delta bcbsp3$ and corresponding complementary strains on Minimal Medium and PDA plates are presented in Supplementary Fig. 5. In revised Fig. 2d, the same amount of spores was cultured to collect EVs from the different strains. After NTA analysis, we found that the amount of EVs released by $\Delta bcpls1$ was largely reduced (Supplementary Fig. 6b). This leads to the reduction of the secreted sRNA. For our analyses, we compared the sRNA amounts between EVs produced from different strains and the WT strain. However, siR17 was not detectable in the WT *B. cinerea* EVs so it is not feasible to present siR17 using real-time PCR. Therefore, we performed small RNA stem-loop RT-PCR and have presented the results in supplementary Fig. 6a. The siR17 result was added to

the small RNA stem-loop RT-PCR. To further confirm our findings, another negative control, siR9, was instead added to the revised manuscript.

In our trans-complementation experiments, fungal EVs were inoculated onto plant leaves and the plant response was detected after two days of inoculation. Under this condition, we did not observe a strong plant response. To test if longer inoculation periods can induce a more robust plant response, we inoculated the plants for a longer time. As shown in Figure 3 below, after four days of inoculation, tiny lesions can be seen on EV-treated plants. This may indicate that the EVs require time to diffuse into plant cells. The fungal cells facilitate the delivery of EVs into the plant apoplastic area by breaking/digesting the plant cell wall. Thus, inoculation of only EVs on the leaf surface requires a longer timeframe to observe the plant response induced by EVs.

Figure 3. Plant response to *B. cinerea* EVs inoculation. EVs were diluted in an equal volume of 2x inoculation buffer and dropped on top leaves. Images were captured at intervals of one day from Day 1 to Day 4. Inoculation buffer served as the negative control.

Figure 3: It is surprising that TyrA23 does not affect *Bc* growth and sporulation. Can the authors explain and show that the treatment does inhibit endocytosis e.g. by FM4-64 tracing? Is it possible that the lesions in figure 3b pictures appear not fully matching the quantitative analysis of *chc2*? And given the much smaller leaves, how informative are results obtained from the *ap2σ* mutant?

Response: As suggested by Reviewer 1, we tried another clathrin-mediated endocytosis-specific inhibitor ES9-17. ES9-17 is a new and specific CME inhibitor, which can bind the CHC protein and block plant endocytosis⁴. However, this inhibitor can also inhibit the growth of *B. cinerea* on the plate. Thus, we decided to use more reliable genetic analysis to address this question by including two more inducible dominant negative mutants INTAM>>RFP-HUB1 and XVE>>AX2. All of these CME mutants exhibited greater resistance to *B. cinerea* infection, which indicates the involvement of CME in plant-*B. cinerea* interaction. To eliminate any potential errors in the measurement of lesion size, we recalculated all relative lesion sizes by comparing them with the sizes of the leaves.

Figure 4: Are all mutants expressing similar levels of AGO1? Is it correct that all experiments were done from BcEV-treated plants but figure 4f shows results from Bc-infected plants?

Response: AGO1 protein levels in different mutants were detected using western blots. As shown in Supplementary Fig. 10, no obvious difference was observed between the plant mutants and treatments we

used in this study. For Fig. 4, all samples were collected from Botrytis-infected plants, not BcEV-treated plants.

Figure 5: Related to figure 1, please show co-localization of BcPSL1-YFP with plant CCV markers.

Response: The colocalization results have been added to Fig. 6c.

Figure 6: It is known that some Arabidopsis proteins have per se an affinity for GFP, therefore, is IgG a suitable control? Please show TEM of CCV-pull-down to confirm the presence of CCVs.

Response: We used an Arabidopsis line that expresses YFP as a negative control in the revised version. We used scanning electron microscopy (SEM) to detect the CCVs attached to the GFP beads. Intact CCVs with typical cage structures can be observed after immunoisolation (Fig. 6b).

Discussion: How do the authors envisage the uptake process of BcEVs into plants and then the binding of Bc small RNAs to plant AGO1? Is BcPSL1 present at CCVs? Is AGO1 present at CCVs? Or are vesicles ruptured to release the Bc small RNAs?

Response: We detected BcPLS1 in CCVs that are isolated from BcPLS1-mCherry-infected CLC1-GFP plants. We found that BcPLS1 can be detected in plant CCVs. Please see Fig 6d in the revised manuscript. We also tried to detect AGO1 in purified CCVs, but our result showed that AGO1 is not present in our isolated CCVs (Figure 4 below).

Figure 4. AGO1 was not detected in CCVs. CCVs were isolated from infected CLC1-GFP plants using immunoisolation. Arabidopsis AGO1 protein was not detected in the CCVs.

Methods: I do not understand the sentences “RNA was extracted from the bead-bound EVs with TRIzol reagent (Invitrogen). Proteins were detected by boiling the bead-bound EVs in SDS-loading buffer before running in a SDS-PAGE gel.” In the context of the AGO1 IPs.

Response: We apologize for the mistake in our earlier version of the text and thank you for bringing it to our attention. It should be “RNA was extracted from the bead-bound AGO1 with TRIzol reagent (Invitrogen). Proteins were detected by boiling the bead-bound AGO1 in the SDS-loading buffer before running in the SDS-PAGE gel” We have corrected it in the revised version.

Reference:

- 1 Weiberg, A. *et al.* Fungal small RNAs suppress plant immunity by hijacking host RNA interference pathways. *Science* **342**, 118-123, doi:10.1126/science.1239705 (2013).
- 2 Wang, H. *et al.* Uptake of oomycete RXLR effectors into host cells by clathrin-mediated endocytosis. *Plant Cell*, doi:10.1093/plcell/koad069 (2023).
- 3 Oliveira-Garcia, E. *et al.* Clathrin-mediated endocytosis facilitates the internalization of Magnaporthe oryzae effectors into rice cells. *Plant Cell*, doi:10.1093/plcell/koad094 (2023).
- 4 Dejonghe, W. *et al.* Disruption of endocytosis through chemical inhibition of clathrin heavy chain function. *Nat Chem Biol* **15**, 641-649, doi:10.1038/s41589-019-0262-1 (2019).
- 5 Fu, Y. *et al.* High-frequency off-target mutagenesis induced by CRISPR-Cas nucleases in human cells. *Nat Biotechnol* **31**, 822-826, doi:10.1038/nbt.2623 (2013).
- 6 Hsu, P. D. *et al.* DNA targeting specificity of RNA-guided Cas9 nucleases. *Nat Biotechnol* **31**, 827-832, doi:10.1038/nbt.2647 (2013).
- 7 Lee, H. C. *et al.* qiRNA is a new type of small interfering RNA induced by DNA damage. *Nature* **459**, 274-277, doi:10.1038/nature08041 (2009).
- 8 Tang, K. F. & Ren, H. The role of dicer in DNA damage repair. *Int J Mol Sci* **13**, 16769-16778, doi:10.3390/ijms131216769 (2012).
- 9 Bonath, F., Domingo-Prim, J., Tarbier, M., Friedländer, M. R. & Visa, N. Next-generation sequencing reveals two populations of damage-induced small RNAs at endogenous DNA double-strand breaks. *Nucleic Acids Res* **46**, 11869-11882, doi:10.1093/nar/gky1107 (2018).
- 10 Lu, W. T. *et al.* Drosha drives the formation of DNA:RNA hybrids around DNA break sites to facilitate DNA repair. *Nat Commun* **9**, 532, doi:10.1038/s41467-018-02893-x (2018).
- 11 Gandía, M., Xu, S., Font, C. & Marcos, J. F. Disruption of ku70 involved in non-homologous end-joining facilitates homologous recombination but increases temperature sensitivity in the phytopathogenic fungus *Penicillium digitatum*. *Fungal Biol* **120**, 317-323, doi:10.1016/j.funbio.2015.11.001 (2016).
- 12 Wang, M. *et al.* Bidirectional cross-kingdom RNAi and fungal uptake of external RNAs confer plant protection. *Nat Plants* **2**, 16151, doi:10.1038/nplants.2016.151 (2016).
- 13 Wang, Q. *et al.* Dicer-like Proteins Regulate the Growth, Conidiation, and Pathogenicity of *Colletotrichum gloeosporioides* from *Hevea brasiliensis*. *Front Microbiol* **8**, 2621, doi:10.3389/fmicb.2017.02621 (2017).
- 14 Yin, C., Zhu, H., Jiang, Y., Shan, Y. & Gong, L. Silencing Dicer-Like Genes Reduces Virulence and sRNA Generation in *Penicillium italicum*, the Cause of Citrus Blue Mold. *Cells* **9**, doi:10.3390/cells9020363 (2020).
- 15 Feng, H. *et al.* Dicer-Like Genes Are Required for H₂O₂ and KCl Stress Responses, Pathogenicity and Small RNA Generation in *Valsa mali*. *Front Microbiol* **8**, 1166, doi:10.3389/fmicb.2017.01166 (2017).
- 16 Werner, B. T., Gaffar, F. Y., Schuemann, J., Biedenkopf, D. & Koch, A. M. RNA-Spray-Mediated Silencing of *Fusarium graminearum* AGO and DCL Genes Improve Barley Disease Resistance. *Front Plant Sci* **11**, 476, doi:10.3389/fpls.2020.00476 (2020).

- 17 Werner, B. T. *et al.* Fusarium graminearum DICER-like-dependent sRNAs are required for the suppression of host immune genes and full virulence. *PLoS One* **16**, e0252365, doi:10.1371/journal.pone.0252365 (2021).
- 18 Duanis-Assaf, D. *et al.* Double-stranded RNA targeting fungal ergosterol biosynthesis pathway controls Botrytis cinerea and postharvest grey mould. *Plant Biotechnol J* **20**, 226-237, doi:10.1111/pbi.13708 (2022).
- 19 Islam, M. T. *et al.* Minicell-based fungal RNAi delivery for sustainable crop protection. *Microb Biotechnol* **14**, 1847-1856, doi:10.1111/1751-7915.13699 (2021).
- 20 Niño-Sánchez, J. *et al.* BioClay™ prolongs RNA interference-mediated crop protection against Botrytis cinerea. *J Integr Plant Biol* **64**, 2187-2198, doi:10.1111/jipb.13353 (2022).
- 21 Haile, Z. M. *et al.* Double-Stranded RNA Targeting Dicer-Like Genes Compromises the Pathogenicity of Plasmopara viticola on Grapevine. *Front Plant Sci* **12**, 667539, doi:10.3389/fpls.2021.667539 (2021).
- 22 Dunker, F. *et al.* Oomycete small RNAs bind to the plant RNA-induced silencing complex for virulence. *Elife* **9**, doi:10.7554/eLife.56096 (2020).
- 23 Shahid, S. *et al.* MicroRNAs from the parasitic plant Cuscuta campestris target host messenger RNAs. *Nature* **553**, 82-85, doi:10.1038/nature25027 (2018).
- 24 Ren, B., Wang, X., Duan, J. & Ma, J. Rhizobial tRNA-derived small RNAs are signal molecules regulating plant nodulation. *Science* **365**, 919-922, doi:10.1126/science.aav8907 (2019).
- 25 Wong-Bajracharya, J. *et al.* The ectomycorrhizal fungus Pisolithus microcarpus encodes a microRNA involved in cross-kingdom gene silencing during symbiosis. *Proc Natl Acad Sci U S A* **119**, doi:10.1073/pnas.2103527119 (2022).
- 26 Ji, H. M. *et al.* Fol-milR1, a pathogenicity factor of Fusarium oxysporum, confers tomato wilt disease resistance by impairing host immune responses. *New Phytol* **232**, 705-718, doi:10.1111/nph.17436 (2021).
- 27 Zhang, X., Henriques, R., Lin, S. S., Niu, Q. W. & Chua, N. H. Agrobacterium-mediated transformation of Arabidopsis thaliana using the floral dip method. *Nat Protoc* **1**, 641-646, doi:10.1038/nprot.2006.97 (2006).
- 28 Cui, C. *et al.* A fungal pathogen deploys a small silencing RNA that attenuates mosquito immunity and facilitates infection. *Nat Commun* **10**, 4298, doi:10.1038/s41467-019-12323-1 (2019).
- 29 Wu, F., Huang, Y., Jiang, W. & Jin, W. Genome-wide identification and validation of tomato-encoded sRNA as the cross-species antifungal factors targeting the virulence genes of Botrytis cinerea. *Front Plant Sci* **14**, 1072181, doi:10.3389/fpls.2023.1072181 (2023).
- 30 Termini, C. M. & Gillette, J. M. Tetraspanins Function as Regulators of Cellular Signaling. *Front Cell Dev Biol* **5**, 34, doi:10.3389/fcell.2017.00034 (2017).
- 31 Gurung, S., Perocheau, D., Touramanidou, L. & Baruteau, J. The exosome journey: from biogenesis to uptake and intracellular signalling. *Cell Commun Signal* **19**, 47, doi:10.1186/s12964-021-00730-1 (2021).
- 32 Rana, S., Yue, S., Stadel, D. & Zöllner, M. Toward tailored exosomes: the exosomal tetraspanin web contributes to target cell selection. *Int J Biochem Cell Biol* **44**, 1574-1584, doi:10.1016/j.biocel.2012.06.018 (2012).

- 33 Laulagnier, K. *et al.* Amyloid precursor protein products concentrate in a subset of exosomes specifically endocytosed by neurons. *Cell Mol Life Sci* **75**, 757-773, doi:10.1007/s00018-017-2664-0 (2018).
- 34 Heusermann, W. *et al.* Exosomes surf on filopodia to enter cells at endocytic hot spots, traffic within endosomes, and are targeted to the ER. *J Cell Biol* **213**, 173-184, doi:10.1083/jcb.201506084 (2016).
- 35 Li, S. *et al.* MicroRNAs inhibit the translation of target mRNAs on the endoplasmic reticulum in Arabidopsis. *Cell* **153**, 562-574, doi:10.1016/j.cell.2013.04.005 (2013).
- 36 Gao, Y. *et al.* Lipid-mediated phase separation of AGO proteins on the ER controls nascent-peptide ubiquitination. *Mol Cell* **82**, 1313-1328.e1318, doi:10.1016/j.molcel.2022.02.035 (2022).

REVIEWERS' COMMENTS

Reviewer #1 (Remarks to the Author):

I congratulate authors for an interesting work and for excellently conducted revisions. I do appreciate in particular the use of genetic lines for their experiments instead of pharmacology.

Just, please, in material and methods, refer systematically the original papers where the lines, mutants and constructs were described originally.

Reviewer #2 (Remarks to the Author):

The authors addressed all my criticism sufficiently. I support publication of this manuscript.

Reviewer #3 (Remarks to the Author):

My earlier comments have been amended within the manuscript, and the additional information has strengthened the findings. I would judge that this is a significant advance in our understanding of cross-kingdom signalling, and supports a role for RNA signalling that has been established in the literature. The work presented supports the conclusions and discussion of the paper. The methodology is sound and the detail given is adequate.

Reviewer #4 (Remarks to the Author):

The authors have significantly improved the manuscript and sufficiently addressed the previously raised concerns. I believe the manuscript is now suitable for publication in Nature Communications.